# GLI transcriptional repression is inert prior to Hedgehog pathway activation

Rachel K. Lex[1], Weiqiang Zhou[2], Zhicheng Ji[2,3], Kristin N. Falkenstein[1], Kaleigh E. Schuler [1], Kathryn E. Windsor[1], Joseph D. Kim [1], Hongkai Ji [2] & Steven A. Vokes [1✉]

The Hedgehog (HH) pathway regulates a spectrum of developmental processes through the transcriptional mediation of GLI proteins. GLI repressors control tissue patterning by preventing sub-threshold activation of HH target genes, presumably even before HH induction, while lack of GLI repression activates most targets. Despite GLI repression being central to HH regulation, it is unknown when it first becomes established in HH-responsive tissues. Here, we investigate whether GLI3 prevents precocious gene expression during limb development. Contrary to current dogma, we find that GLI3 is inert prior to HH signaling. While GLI3 binds to most targets, loss of *Gli3* does not increase target gene expression, enhancer acetylation or accessibility, as it does post-HH signaling. Furthermore, GLI repression is established independently of HH signaling, but after its onset. Collectively, these surprising results challenge current GLI pre-patterning models and demonstrate that GLI repression is not a default state for the HH pathway.

[1] Department of Molecular Biosciences, University of Texas at Austin, Austin, TX, USA. [2] Department of Biostatistics, Johns Hopkins Bloomberg School of Public Health, Baltimore, MD, USA. [3]Present address: Department of Biostatistics and Bioinformatics, Duke University School of Medicine, Durham, NC, USA. ✉email: svokes@austin.utexas.edu

The Hedgehog (HH) signaling pathway is one of the major developmental regulators of tissue-specific development and differentiation. GLI proteins mediate transcriptional responses to the pathway in a strikingly cell-specific fashion. In the absence of HH pathway activation, GLI transcriptional repressors (GLI-R) prevent the activation of HH target genes, while upon exposure to HH, GLI proteins undergo alternative post-translational processing into transcriptional activators (GLI-A)[1–5]. The processing of GLI-A and GLI-R are both directly dependent on processes localized within the primary cilium and defects in ciliary components lead to characteristic misregulation of HH target genes in a large group of birth defects termed ciliopathies[6]. While GLI-A directly activates a subset of HH targets, most genes are 'de-repressed' by the loss of GLI-R alone; they do not require GLI-A-mediated activation for their expression. The importance of de-repression is exemplified in the developing limb where the phenotype of Sonic Hedgehog (Shh) null limb buds is dramatically improved in Shh;Gli3 mutants[7–9]. In this context, the loss of GLI3-mediated repression, even in the absence of pathway activation, is sufficient to restore expression of most GLI target genes and many aspects of limb growth and patterning.

GLI3 represses transcription, at least in part, by epigenetically regulating a subset of its own enhancers. Its properties include reduced enrichment levels of the active enhancer mark H3K27ac and reduced chromatin accessibility at a subset of HH-responsive GLI3 binding regions (GBRs), termed HH-responsive GBRs, that likely mediate the majority of HH-specific transcription[10]. As a regulator of tissue patterning, GLI3-R spatially and temporally restricts expression of HH targets, preventing sub-threshold activation of the pathway in HH-responsive tissues[1,11,12]. Consequently, GLI transcriptional repression has primarily been studied in tissues with ongoing HH signaling. Although not experimentally addressed, it is widely assumed that GLI3-R also regulates the expression of its target genes prior to the initiation of HH signaling[8,9,13–16]. Before pathway activation, GLI3 has been proposed to have a pre-patterning role in the very early limb by repressing Hand2, thus limiting its expression to the posterior limb bud, where it is required to activate Shh expression, thereby establishing anterior-posterior polarity[8,9,13–17]. Curiously, Hand2 and Gli3 are co-expressed in early limb buds, prior to their segregation into distinct posterior and anterior domains[14], raising the question of whether GLI3-R is capable of repressing Hand2 at this time.

We initially hypothesized that GLI3-mediated repression would be important prior to HH expression for preventing premature activation of target genes by reducing H3K27ac enrichment at enhancers. Despite GLI3 binding to a majority of the same sites it occupies in the post-HH limb bud, we found that it does not regulate activation of its enhancers or their chromatin accessibility until after the activation of the HH pathway and does so independently of HH signaling. In addition, GLI target genes are not upregulated in limb buds lacking Gli3 as they are after the initiation of HH signaling, even though a subset of these genes appear competent to be repressed at this stage. Contrary to our initial hypothesis, we conclude that the GLI3-R isoform is transcriptionally inert in early limb buds, a finding that is incompatible with the pre-patterning model for limb polarity[8,9,13–16]. Interestingly, most genes that are later repressed by GLI3 are expressed in pre-HH limb buds, suggesting that rather than preventing their activation, GLI3 repression later regulates their spatial expression after HH signaling initiates. Overall, this work demonstrates that GLI3-mediated repression of target genes is not a default state for the Hedgehog pathway; instead GLI repression is established during limb development at a time point after HH induction.

## Results

**GLI3-R is abundant and binds chromatin prior to HH signaling**. To understand if GLI repression is established prior to HH signaling, we first defined when the pathway is activated during limb development. The earliest detection of the canonical HH target gene Gli1 was at 24 somites (24 S), where 58% of embryos at this stage had detectable Gli1 expression, while Shh was not detected until 25 S (86%) (Fig. 1a, b; Supplementary Fig. 1a–c). As 24 S was the earliest detection of Gli1 expression, we defined that stage as the onset of HH signaling and the "pre-HH" window as 21–23 S, corresponding to embryonic day 9.25 (E9.25), a stage slightly earlier than previous reports (Fig. 1)[18–20].

Confirming previous findings, GLI3-R was expressed at comparable levels at both pre- and post-HH stages (Fig. 1c, Supplementary Fig. 1d, Source Data)[14]. We then compared endogenous GLI3$^{FLAG}$ binding using CUT&RUN[21] in pre-HH (E9.25, 21-23 S) and anterior E10.5 (32-35 S) limb buds, a post-HH time frame when HH signaling is firmly established in the posterior limb, but before there are morphological changes in $Shh^{-/-}$ limb buds[22]. In pre-HH limb buds, GLI3 bound to most regions that were bound in post-HH limb buds (82%; Fig. 1d, Supplementary Data 1). Previously, we identified a group of HH-responsive GLI3 binding regions that have reduced H3K27ac in $Shh^{-/-}$ limb buds (constitutive GLI repression)[10] (Supplementary Fig. 1g–e). Since this subset of GBRs seems to regulate most HH-responsive gene expression, these regions might be especially important to repress before HH signaling to prevent premature activation of enhancers. While the majority of HH-responsive GBRs (70%) were bound by GLI3 prior to HH induction, nearly a third were not bound, suggesting that GLI3 does not initially regulate a subset of HH-responsive GBRs (Fig. 1e).

**GLI3 preferentially binds to poised, accessible enhancers**. Because GLI3 bound to only a portion of regions in the pre-HH limb, we hypothesized that GLI3 may preferentially bind to poised enhancers. We defined poised enhancers as ATAC-seq accessible and enriched for either H3K4me1 or H3K4me2, where H3K4me1 is enriched at promoter-proximal and distal regions, while H3K4me2 is more commonly found promoter proximally[23–25]. Most HH-responsive GBRs that were accessible and enriched for H3K4me2 by E10.5 were already accessible (89%) and enriched for H3K4me2 (98%) in the early limb (Supplementary Fig. 1h, i, Supplementary Data 2). In contrast, only 65% of the HH-responsive GBRs with H3K4me1 enrichment at E10.5 were also enriched for H3K4me1 by E9.25 (Supplementary Fig. 1i). We then asked if GLI3 preferentially bound to accessible, poised regions. Consistent with this scenario, most regions bound by GLI3 in pre-HH limbs were accessible (93%) and enriched for poised enhancer modifications (75%), including a defined limb-specific distal enhancer for Ptch1[26] (Fig. 1f–h; Supplementary Fig. 1j). While nearly half of the regions not yet bound by GLI3 in the early limb overlapped with called ATAC-seq peaks (Supplementary Fig. 1j), they were less accessible than GLI bound enhancers (Fig. 1g). Unbound HH-responsive GBRs also generally lacked enrichment of poised enhancer marks, H3K4me1 and H3K4me2 (Fig. 1g, i; Supplementary Fig. 1j). This is exemplified by the distal limb enhancer, GRE1 that helps regulate the HH target Gremlin, which is among the inaccessible regions that lack H3K4me1 enrichment and are not bound by GLI3 at E9.25 (Fig. 1i). This finding is consistent with previous reports demonstrating that GRE1 does not have enhancer activity until E10 (31–32 S)[27,28]. As many distal regions lacked H3K4me1 enrichment prior to HH induction, we observed a slight preference for GLI3 to be bound to promoter proximal regions (Supplementary Fig. 1k). We conclude that GLI3 preferentially

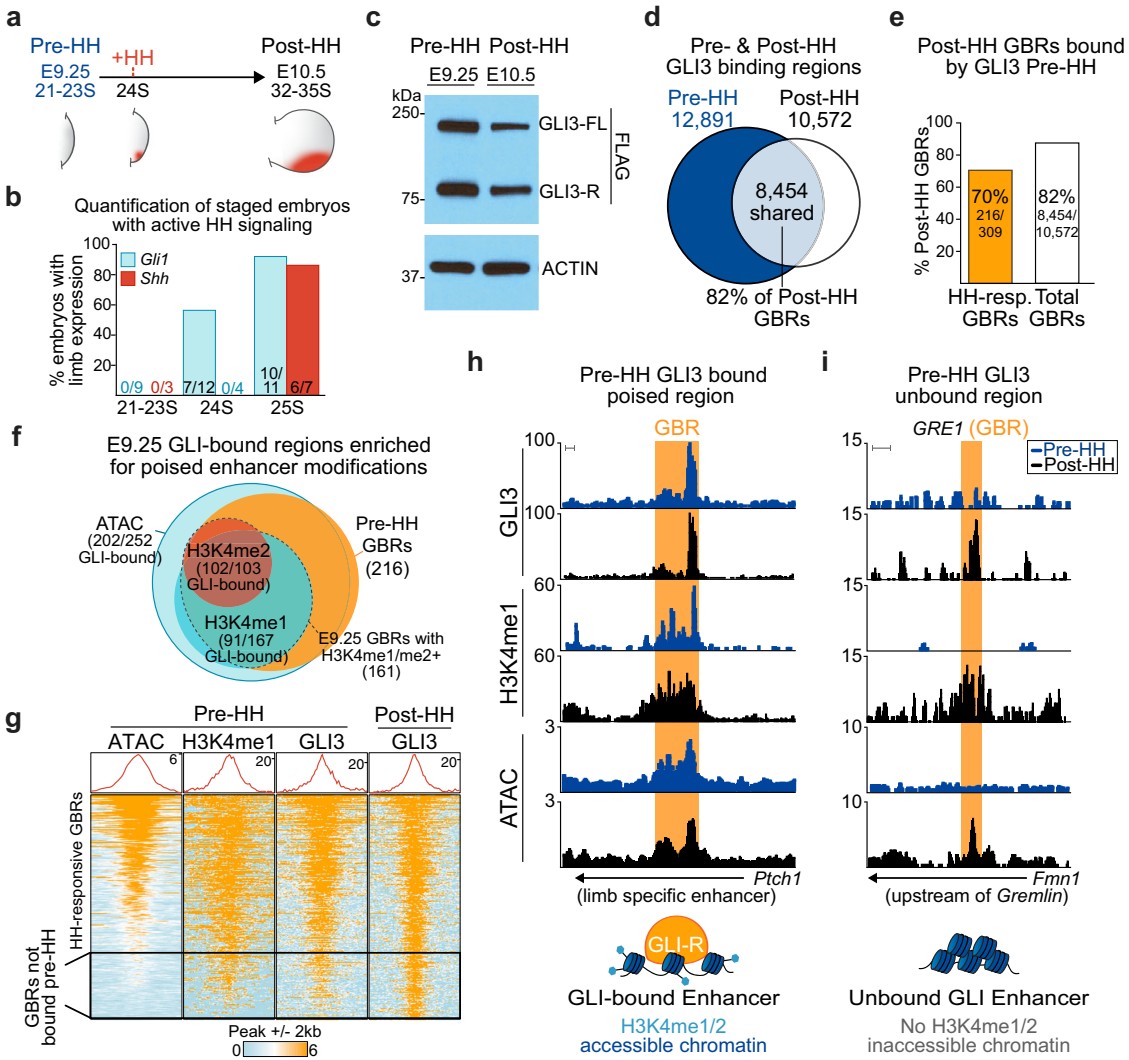

**Fig. 1 GLI3 binds to poised, accessible chromatin prior to HH signaling. a** Schematic for HH induction timeline during limb development. **b** Quantification of 21–25 S embryos with expression of *Gli1* and *Shh* assayed by in situ hybridization. **c** Representative western blot (*n* = 3) of endogenous GLI3[FLAG] protein in the limb bud pre-(21–23 S) and post-HH signaling (32–35 S). **d** Venn diagram of all pre- and post-HH identified GLI3 CUT&RUN called peaks. **e** Percentage of E10.5 GLI3-bound regions also bound at E9.25, before HH signaling, for all E10.5 identified GBRs and E10.5 HH-responsive GBRs. **f** Venn diagram of HH-responsive GBRs enriched for poised enhancer marks and bound by GLI3 at E9.25. Poised enhancer modifications identified by H3K4me1 CUT&Tag (*n* = 3), H3K4me2 ChIP-seq (*n* = 2), ATAC-seq (*n* = 2). **g** Heatmap of GLI3 enrichment pre and post-HH signaling and enrichment of poised enhancer marks pre-HH signaling. Many regions lacking GLI3 also lack enrichment of poised marks. **h** Example of a HH-responsive GBR (orange shading) that is poised, accessible, and bound by GLI3 prior to HH signaling at a limb-specific *Ptch1* enhancer[26]. **i** Example of a validated HH-responsive GBR GRE1, that regulates the GLI3 target *Gremlin*[27], that is inaccessible, lacks H3K4me1 and GLI3 binding at E9.25. Scale bars = 1 kb. Source data are provided as a Source Data file.

binds to poised enhancers, which may provide tissue-specific control of repression to prevent precocious expression.

**GBRs are HDAC-bound with low H3K27ac before HH activation.** If GLI3 represses enhancers in the early limb as it does at E10.5, then HH-responsive GBRs, which have reduced H3K27ac in E10.5 *Shh*$^{-/-}$ limb buds, should have reduced acetylation at E9.25 before HH signaling initiates. In agreement with this scenario, there was a significant reduction in H3K27ac enrichment at HH-responsive GBRs in pre-HH compared to post-HH limb buds (Fig. 2a–c, Supplementary Data 2). In addition, only 39% (121/309) of HH-responsive GBRs have called H3K27ac peaks in the pre-HH limb, of which, 88% (107/121) are bound by GLI3 in the pre-HH limb, with only 14 GBRs not bound by GLI3 being enriched for H3K27ac at this time. In contrast to HH-responsive GBRs, we previously found most GBRs remain stably acetylated

in both the presence and absence of HH signaling at E10.5[10]. Consistent with this, 86% (6385/7382; Supplementary Fig. 1e) of all E10.5 GBRs that are acetylated in E10.5 limb buds, have called H3K27ac peaks prior to HH induction. The overall reduction in H3K27ac enrichment specifically at HH-responsive GBRs in the early limb bud, supports the possibility that GLI3 could be actively repressing enhancers at this time to prevent premature activation.

Previously, we found that GLI-mediated repression is facilitated by HDACs that regulate H3K27ac enrichment at GBRs[10]. Pre-HH limb buds had similar levels of HDAC1 and HDAC2, with many GBRs enriched for both HDACs (Fig. 2d, e, Supplementary Data 1). In post-HH limb buds, most of these regions continue to be bound by HDAC1 while HDAC2 binding is greatly reduced (Fig. 2f). Overall, the presence of HDACs at most GBRs prior to HH is consistent with a model in which GLI3,

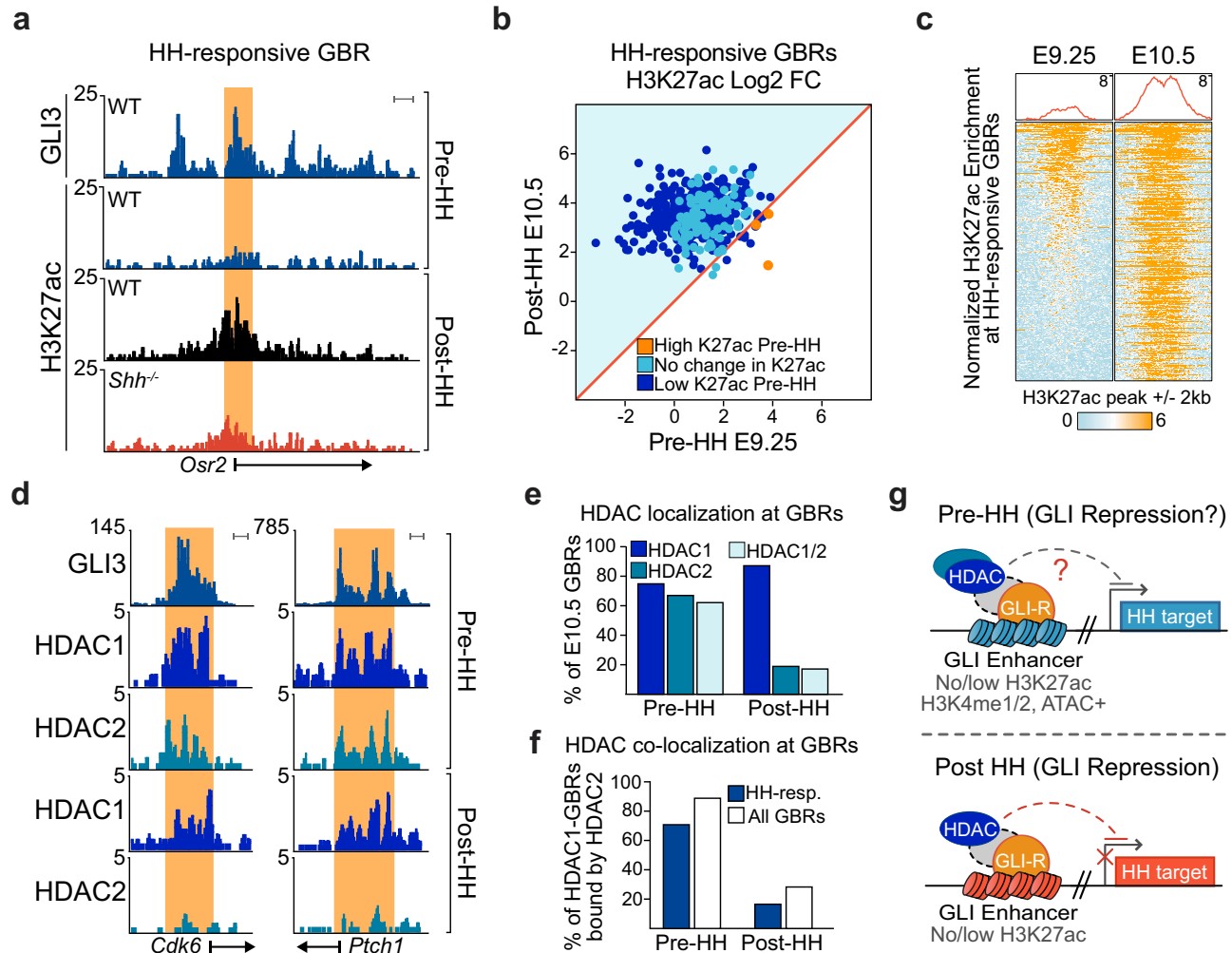

**Fig. 2 GLI3 binding regions have reduced enrichment of H3K27ac and are enriched with HDACs in pre-HH limb buds. a** H3K27ac ChIP-seq tracks showing a E9.25 (21-23 S) GLI3-bound HH-responsive GBR with reduced acetylation in E9.25 WT limb buds compared to E10.5 WT limbs[10] (note that the levels of enrichment are comparable to E10.5 $Shh^{-/-}$ limbs). **b** Scatter plot of H3K27ac enrichment at HH-responsive GBRs in pre- and post-HH[10] WT limbs. HH-responsive GBRs with significant reductions (FDR < 0.05) in H3K27ac at E9.25 compared to E10.5, are denoted in dark blue, while the 3 GBRs with significantly higher H3K27ac in E9.25 limbs are in orange (FDR < 0.05 $n = 2$). **c** Heatmap of H3K27ac ChIP-seq indicating relative H3K27ac enrichment at HH responsive GBRs in pre- and post-HH[10] limbs ($n = 2$). **d** GLI3, HDAC1 and HDAC2 CUT&RUN tracks at E9.25 GLI3-bound regions, pre- and post-HH. **e** HDAC localization at E10.5 bound GBRs. Prior to HH signaling, most regions are enriched for both HDAC1 and HDAC2, while after, most GBRs are enriched for only HDAC1 (E9.25 HDAC1/2 $n = 2$; E10.5 HDAC1/2 $n = 3$). **f** Initially most HDAC1-bound regions are also enriched for HDAC2, while after HH-signaling, fewer HDAC1-bound regions are also enriched for HDAC2. **g** Prior to HH signaling, Gli3-bound regions are enriched for poised enhancer marks with lower levels of H3K27ac and HDAC enrichment. These are comparable to post-HH GLI3 enhancers where GLI3 actively represses targets. Orange shading in tracks indicates HH-responsive GBRs defined in Supplementary Fig. 1e. Scale bars = 1 kb.

together with an HDAC-containing repression complex, could be repressing enhancers to prevent premature activation of target genes (Fig. 2g).

**Loss of *Gli3* does not prematurely activate GLI3 enhancers.** We next tested the hypothesis that the reduction in acetylation at GBRs prior to HH signaling was due to GLI3-mediated reduction in the levels of H3K27ac that prevented premature activation of enhancers (Fig. 2g). We examined H3K27ac enrichment in WT and *Gli3*$^{-/-}$ pre-HH limb buds (Fig. 3a; Supplementary Data 2), predicting that loss of GLI3-R at this time would result in increased acetylation at GBRs, as it does in E10.5 post-HH limbs[10] (Fig. 3b). Surprisingly, and contrary to our hypothesis, there was no significant increase in H3K27ac enrichment in *Gli3*$^{-/-}$ limb buds prior to HH induction (Fig. 3b–f). Since GLI3-mediated repression of *Hand2* in the early limb is a key

component of the pre-patterning model of anterior-posterior polarity[8,14,16,29,30], we examined two GBRs located 10 kb and 85 kb downstream of *Hand2* that mediate GLI repression[15], but did not observe any increases in H3K27ac (Fig. 3e, Supplementary Fig. 2a, Supplementary Data 2). Additional GBRs around established GLI3 target genes also did not show increased H3K27ac enrichment with loss of *Gli3*. This was evident at genes such as *Hoxd12* which lacked WT E9.25 H3K27ac at GLI-bound regions and remained unchanged in *Gli3*$^{-/-}$ limbs, or at genes like *Cdk6*[31], which had E9.25 acetylation, but did not gain further enrichment with loss of *Gli3* (Fig. 3e). We next investigated other genes like *Cdk6*, as we considered the possibility that many enhancers, which have low levels of H3K27ac, may not be competent for activation at this stage and would therefore not have increased enrichment of H3K27ac upon loss of *Gli3*. To address this, we examined three outlier HH-responsive GBRs that contained significantly higher levels of H3K27ac in pre-HH than

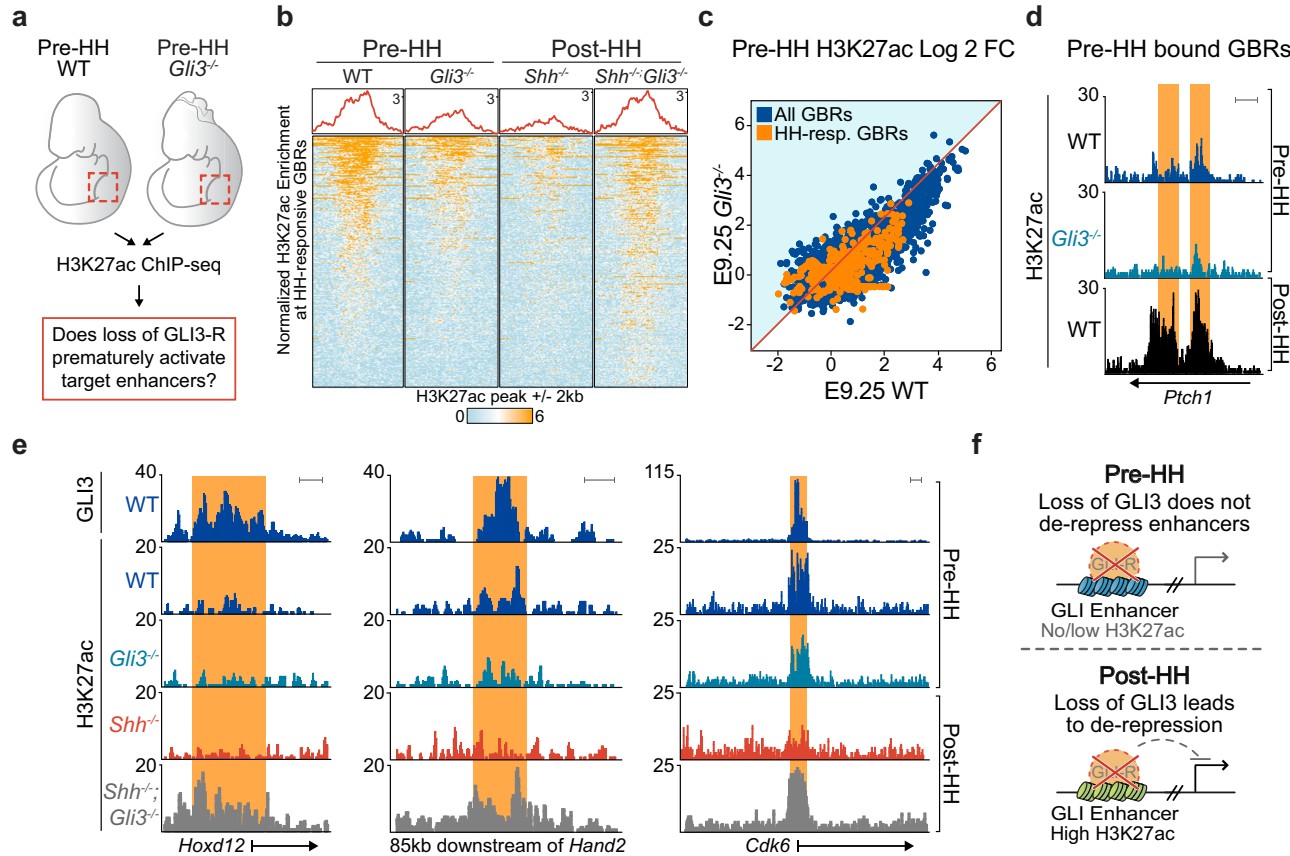

**Fig. 3 Loss of *Gli3* does not result in premature de-repression of enhancers. a** Schematic for testing whether loss of *Gli3* prematurely increases H3K27ac levels through GLI3 de-repression. **b** Heatmap of H3K27ac enrichment at HH-responsive GBRs in pre-HH (E9.25, 21–23 S) and post-HH (E10.5, 33-34 S) limb buds, with loss of *Gli3*. At E10.5, *Shh⁻/⁻* limbs have reduced H3K27ac due to GLI3 repression. Loss of *Gli3* in *Shh⁻/⁻;Gli3⁻/⁻* results in de-repression of target enhancers and increased acetylation. This change is not observed prior to HH signaling with loss of *Gli3* compared to WT limb buds. **c** Scatterplot of H3K27ac enrichment in pre-HH WT vs. *Gli3⁻/⁻* limbs shows no increase in acetylation with loss of *Gli3*. **d** H3K27ac ChIP-seq tracks at a pre-HH GLI3-bound HH-responsive GBR with low H3K27ac in E9.25 WT limb buds that does not increase acetylation levels in E9.25 *Gli3⁻/⁻*. **e** Examples of regions, bound by GLI3 at E9.25, that do not increase H3K27ac levels with loss of *Gli3* in pre-HH limb buds as they do in *Shh⁻/⁻;Gli3⁻/⁻* post-HH limbs.
**f** Schematic depicting loss of *Gli3* in pre-HH limb buds does not lead to de-repression of target enhancers as it does after HH induction at E10.5. Orange shading on tracks indicates HH-responsive GBRs defined in Supplementary Fig. 1e. Scale bars = 1 kb.

post-HH limb buds (Fig. 2b) to see if the acetylation levels might further increase in the absence of GLI3 repression. Despite GLI3 binding at these regions, there was no increase in H3K27ac enrichment with loss of *Gli3* before HH signaling (Supplementary Fig. 2b–d). Overall, this surprising result indicates that despite the presence of GLI3 and HDACs, GLI repression does not regulate H3K27ac enrichment at enhancers in pre-HH limb buds (Fig. 3f).

**Loss of *Gli3* does not prematurely activate GLI3 target genes.** To determine whether the lack of enhancer activation in E9.25, pre-HH *Gli3⁻/⁻* limbs corresponded with a lack of de-repression of GLI target genes, we performed RNA-seq on pre-HH WT and *Gli3⁻/⁻* limbs, as well as the anterior halves (no HH; GLI-repressed domain) of post-HH (32-35 S) WT and *Gli3⁻/⁻* limbs (Fig. 4a–d, Supplementary Data 3). In post-HH *Gli3⁻/⁻* limbs, there were 159 significantly upregulated genes (FDR < 0.05), including well known signature HH target genes such as *Ptch1*, *Hand2*, *Hoxd13* and *Grem1* (Fig. 4c, d)[7]. However, in agreement with the lack of GLI3-regulated enhancer activity, very few genes (9) were significantly upregulated with loss of *Gli3* in the early limb (Fig. 4b). These genes included *Foxf1* and *Osr1*, which are regulated by HH signaling in several mesodermal tissues but, with the exception of *Gpx6*, are not upregulated in anterior E10.5 *Gli3⁻/⁻* limbs[32–35]. This suggests that rather than being a product

of ongoing GLI3 repression, these upregulated genes may represent outliers from contaminating flank tissue from the dissection. Alternatively, these could represent residual mRNAs from the lateral plate mesoderm prior to their epithelial to mesenchymal transition at the onset of limb initiation within the last ~8 h[36]. Consistent with the former possibility, *Foxf1* is present in the lateral plate mesoderm but is not detected in either WT or *Gli3⁻/⁻* pre-HH limb buds (Fig. 4d). Similarly, *Ptch1*, a signature HH target gene was detected in the neural tube, but was also not detected in the early limb buds in WT or *Gli3⁻/⁻* embryos (Fig. 4d). In support of the latter possibility, pre-HH upregulated mRNAs are reported to have an average half-life of 11.2 h in differentiating ES cells (Supplementary Fig. 3a)[37], and also have lower levels of retained introns than post-HH upregulated *Gli3⁻/⁻* genes, suggesting that some of these transcripts may represent older mRNAs, potentially regulated by GLI3 in the lateral plate mesoderm, that are not currently being transcribed in the limb (Supplementary Fig. 3b, c). This would be consistent with recent findings that the early limb transcriptome is a continuation of that observed in trunk progenitors[38]. We conclude that GLI3 is unlikely to be engaged in ongoing transcriptional repression in the early limb bud.

As *Hand2* is the best-characterized target of GLI3 repression, we examined *Hand2* expression in pre-HH and early post-HH

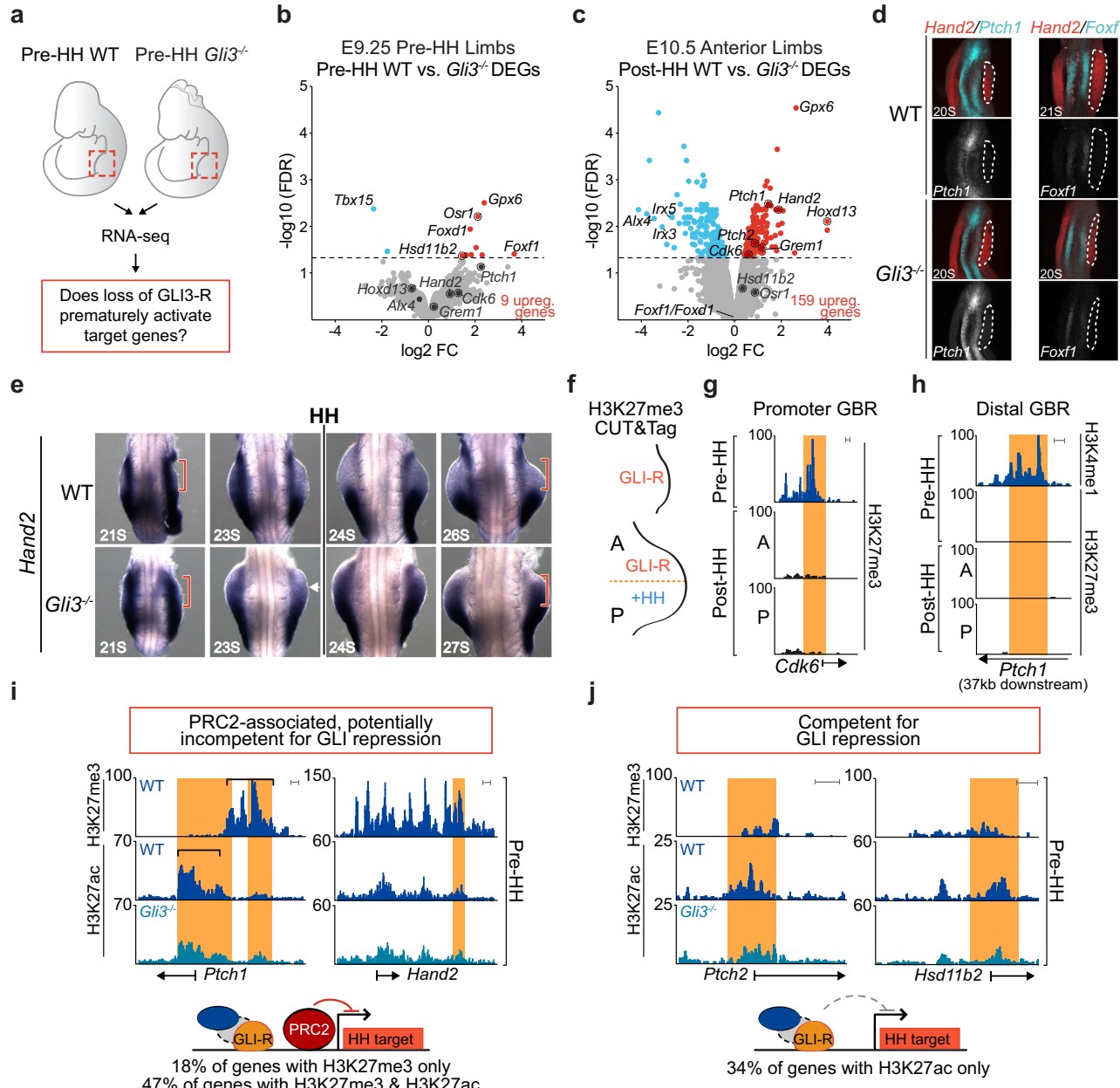

**Fig. 4 Most genes are not de-repressed in *Gli3* mutants prior to HH induction. a** Schematic for testing whether loss of *Gli3* can prematurely activate target genes through GLI3 de-repression. **b**, **c** Volcano plot of differentially expressed genes (DEGs) detected in RNA-seq between WT and *Gli3*$^{-/-}$ limbs E9.25 (21–23 S) (**b**) and E10.5 (32–35 S) anterior limb buds (**c**). **d** Fluorescent in situ hybridization showing maximum intensity projections for *Hand2, Foxf1* and *Ptch1* in E9.25 WT and *Gli3*$^{-/-}$ limb buds indicating the absence of detectable *Foxf1* and *Ptch1* in pre-HH limb buds. Dashed white lines outline the limb bud region. **e** In situ hybridization for the GLI3 target *Hand2*, in WT and *Gli3*$^{-/-}$ limbs pre- (21, 23 S) and immediately post-HH induction (24–27 S; *n* = 3). Note that *Hand2* is expressed almost uniformly across the limb bud in 21 S embryos in both WT and *Gli3*$^{-/-}$ limbs. At 23 S, slight reduction in anterior expression of *Hand2* is observed in both WT and *Gli3*$^{-/-}$ limbs (white arrowhead). The red brackets denote anterior limb. Images acquired on a Zeiss LSM 710 Confocal visualized using maximal intensity projections with pseudo-coloring for merged images. **f** Schematic identifying H3K27me3 enriched regions in E9.25, E10.5 anterior (GLI3-repressed), and E10.5 posterior (HH signaling, loss of GLI-R) limb buds using CUT&Tag. **g–j** Orange shading in tracks indicates HH-responsive GBRs defined in Supplementary Fig. 1e. **g** Example of a HH-responsive GBR at a HH target gene promoter with H3K27me3 enrichment specifically at E9.25 but not E10.5. **h** Representative distal limb-specific HH-responsive GBR downstream of *Ptch1* with no H3K27me3 enrichment in E9.25 or E10.5 limb buds. **i** Examples of HH target genes associated with polycomb repression that could potentially be incompetent for GLI3 repression. **j** Examples of genes that are bound by GLI3 at E9.25, contain H3K27ac enrichment and lack H3K27me3 enrichment, suggesting they are competent for GLI3 repression but are not de-repressed in the absence of *Gli3*. For **f–j** see Supplementary Data 1. Scale bars for tracks indicate 1 kb.

WT and *Gli3*$^{-/-}$ embryos to look for subtle changes that might not be detected by RNA-seq. *Hand2* is initially expressed throughout anterior WT limb buds (Fig. 4d, e) similar to *Gli3*$^{-/-}$ limb buds (20–23 S) (*n* = 7). At early post-HH stages (24–26 S), *Hand2* expression is variable in WT and *Gli3*$^{-/-}$ limb buds, with most WT limb buds expressing anterior *Hand2* at levels comparable to those seen posteriorly. However, comparably lower levels of anterior *Hand2* expression were present even in some *Gli3*$^{-/-}$ limb buds (Fig. 4e, white arrow). Posterior restriction of *Hand2* is not evident until 26 S and is still variable

at that time (Fig. 4e, Supplementary Fig. 3e). Although not quantitative, these findings are consistent with previous reports that noted early co-expression of *Gli3* and *Hand2* at these stages[14]. We conclude that GLI3 repression of *Hand2* is absent prior to the activation of HH signaling and suggests that GLI3 repression is first activated around 26 S, shortly after the initiation of HH signaling (Supplementary Fig. 1-c).

**GLI3 repressor is inert in early limb buds.** Our results thus far indicated that GLI repression is not active in the early limb bud. This could either be caused by inert GLI repression complexes or by target genes that were incapable of being activated. To understand whether an alternative form of repression might compensate for the lack of GLI3 repression, we asked if the Polycomb Repressive Complex 2 (PRC2) might broadly regulate HH targets in pre-HH limb buds. We predicted a set of 74 high confidence GLI target genes by identifying genes downregulated in *Shh*$^{-/-}$ limbs that were near HH-responsive GBRs (Supplementary Data 4)[7]. We then performed CUT&Tag[39] for H3K27me3, a modification indicative of PRC2 repression in pre-HH limbs and anterior and posterior post-HH limb buds (Fig. 4f–h; Supplementary Data 1)[40]. There was little H3K27me3 enrichment at target gene promoters after HH induction at E10.5, with the notable exceptions of *Ptch1* and *Gli1* in anterior E10.5 limbs, consistent with previous reports[10] (Fig. 4h, I; Supplementary Fig. 4a-d; Supplementary Data 4). However, most (81%) putative HH target gene promoters were enriched for H3K27me3 in pre-HH E9.25 limb buds (Supplementary Fig. 4a). This H3K27me3 enrichment appears to resolve over limb development, as H3K27me3 enrichment is greatly reduced or lost at most genes in post-HH limb buds, with only 21% remaining trimethylated in anterior E10.5 limb buds (Fig. 4h; Supplementary Fig. 4a–d). Most HH-responsive GBRs are located distally and lacked H3K27me3 enrichment, while the few GBRs enriched for H3K27me3, were generally located proximal to promoters, suggesting PRC2 repression is likely independent of GLI regulation (Fig. 4i; Supplementary Fig. 4b).

Interestingly, there were high levels of H3K27me3 at the *Hand2* locus in the early limb (Supplementary Fig. 4f), which as noted above is widely expressed throughout the limb at this time. *Hand2* also was enriched for H3K27ac, which is mutually exclusive with H3K27me3[41], suggesting that PRC2 is associated with *Hand2* in a subset of cells that presumably do not express *Hand2* in pre-HH limb buds (Supplementary Fig. 4f). Similar to *Hand2*, nearly half of predicted HH target genes were enriched for both H3K27ac and H3K27me3, which was also true of the few E10.5 genes enriched for H3K27me3 (Supplementary Figs. 4d, 7c; Supplementary Data 4). At E9.25, most genes have high levels of H3K27me3 and low levels of H3K27ac, while at E10.5, when trimethylation is resolving, most genes have increased levels of H3K27ac with low levels of H3K27me3, consistent with increases in their expression levels at E10.5 (Supplementary Fig. 4d–g). For other genes enriched for both marks, like *Ptch1*, the distribution of these modifications is offset, supporting the potential for these mutually exclusive marks to be present at the same locus (Fig. 4i, black brackets). The dual enrichment of H3K27ac and H3K27me3 has been proposed to enable fast, precise induction of these genes upon presence of relevant stimuli[42]. A smaller population of genes (18%), are highly enriched for H3K27me3 and completely lack acetylation at E9.25, suggesting they may not be competent for de-repression upon loss of *Gli3* at this stage (Fig. 4i; Supplementary Fig. 7c). Importantly, a third of genes are bound by GLI3 at E9.25, but have no H3K27me3 enrichment and are likely competent for de-repression upon the loss of *Gli3* (Fig. 4j; Supplementary Fig. 8a). However, as we did not observe

increased acetylation in E9.25 *Gli3*$^{-/-}$ limbs at these regions as we do in E10.5 *Gli3*$^{-/-}$ limb buds, we conclude that GLI3 repressor is inert in the early limb bud (Fig. 4j).

**GLI-dependent chromatin compaction occurs after HH induction.** We previously noted that there was reduced chromatin accessibility in posterior E10.5 *Shh*$^{-/-}$ (constitutive GLI repression) limb buds at HH-responsive GBRs[10], compared to posterior WT limbs which have active HH signaling and lack GLI repression. To determine if this was caused by GLI repression, as opposed to a lack of GLI activator, we performed ATAC-seq in posterior E10.5 *Shh*$^{-/-}$;*Gli3*$^{-/-}$ limb buds. Loss of *Gli3* in the absence of *Shh* resulted in accessible chromatin at HH-responsive GBRs similar to posterior WT controls, confirming that chromatin compaction is GLI3 repressor-dependent (Fig. 5a). To determine the onset of GLI3-mediated chromatin compaction, we examined chromatin accessibility using ATAC-seq at HH-responsive GBRs in E9.25 pre-HH limb buds (21–23 S), posterior WT and posterior *Shh*$^{-/-}$ limbs at E10.5 (35 S) when HH signaling is firmly established and an intermediate timepoint, E10 (28–30 S), 8–12 h after the initiation of HH signaling. We first examined *Hand2*, whose expression becomes posteriorly restricted in the limb around 26 S (Supplementary Fig. 3e). We anticipated that GLI3 repression should be established by E10 (28–30 S), and predicted that chromatin at HH-responsive GBRs in *Shh*$^{-/-}$ limbs at this stage would be more compact compared to E10 (28–30 S) WT controls, in addition to pre-HH WT limbs, which contain accessible chromatin and lack active GLI repression. Alternatively, if GLI-repression was still in the process of being established, the accessibility of HH-responsive GBRs would be similar to that of pre-HH and E10 (28–30 S) WT limbs. Consistent with an absence of GLI repression, the chromatin at HH-responsive GBRs in posterior E10 (28–30 S) *Shh*$^{-/-}$ limb buds was not yet compacted. In fact, chromatin was significantly more accessible in E10 (28–30 S) *Shh*$^{-/-}$ limb buds compared to both E9.25 (21–23 S) WT limbs and posterior E10 (28–30 S) WT limbs, which have active HH signaling and little to no nuclear GLI3-R[10] (Fig. 5a–d). This surprising result suggests that GLI repression has not been completely established in this population at this stage in development. We next examined individual GBRs to determine if there were any regions with less accessible chromatin in E10 (28–30 S) *Shh*$^{-/-}$ limb buds, indicative of initial targets of GLI3 repression. While most regions were not significantly reduced, several regions trended toward having reduced accessibility at E10 (28–30 S) in *Shh*$^{-/-}$ compared to WT, such as HH-responsive GBRs near HH targets *Ptch1* and *Hhip* (Fig. 5e–g, blue datapoints). However, this reduction did not become significant until E10.5 (Fig. 5e–g, red datapoints). Only a few HH-responsive GBRs had significantly reduced accessibility in E10 (28–30 S) *Shh*$^{-/-}$ limb buds including GBRs around *Cdh11* and *Mllt3*, which were also significantly reduced at E10.5 (35 S) (Fig. 5e, f, h, i; Supplementary Data 5). Since *Shh*$^{-/-}$ limbs have constitutive GLI repression, the lack of GLI3-dependent chromatin compaction in E10 (28–30 S) *Shh*$^{-/-}$ limbs supports that GLI2-R is unlikely to functionally compensate for the lack of GLI3 repression in early limbs (see discussion). Collectively these results support that GLI repression is not fully established even up to 12 h after HH signaling would have normally become active (Fig. 5j).

**GLI3 ciliary changes coincide with onset of GLI repression.** Prior to entering the nucleus, GLI proteins traffic through the primary cilium and are subsequently modified into their truncated repressor form. As GLI3 repression does not initiate until after the induction of HH signaling, we asked if there was a

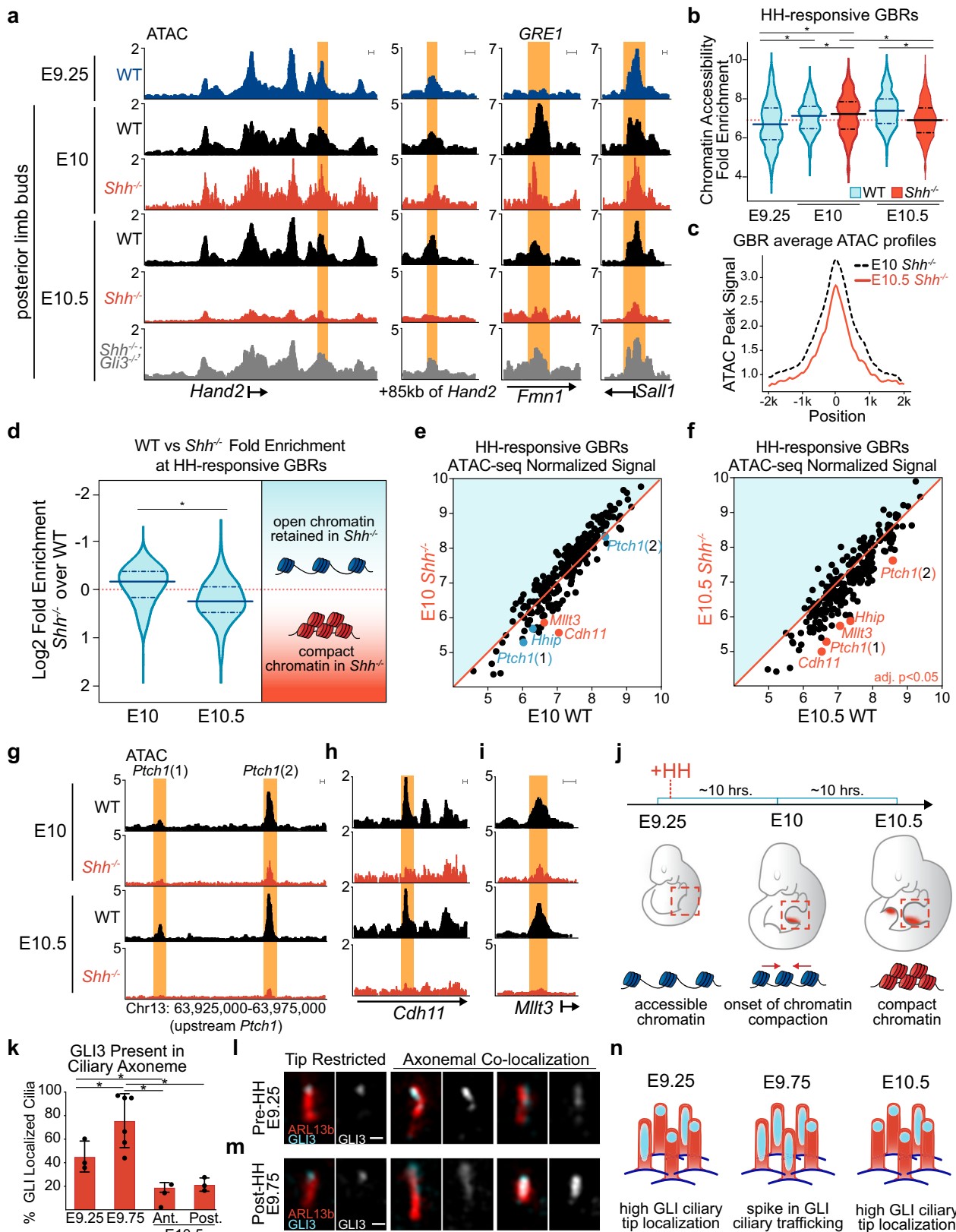

change in the ciliary trafficking of GLI proteins that could signify changes in the processing of GLI-R. In pre-HH (E9.25, 21–23 S) limb buds, endogenous GLI3[FLAG] was localized to ~80% of cilia (Supplementary Fig. 5a, b, e). There were similarly high trends in GLI3, as well as GLI2, ciliary co-localization in pre-HH WT and $Shh^{-/-}$ limb buds, where they were primarily localized to the ciliary tip (Supplementary Fig. 5a–e). This indicates that this enrichment is not due to undetectable levels of SHH that might still promote GLI activation and, as GLI-R is not localized to the cilia[43], it suggests that pre-HH ciliary localization is enriched with full-length, unprocessed GLI protein (Supplementary Fig. 5a–e)[43,44]. We then examined GLI ciliary localization shortly

**Fig. 5 GLI3-dependent chromatin compaction occurs after HH induction. a** ATAC-seq tracks showing examples of HH-responsive GBRs that are compacted in posterior E10.5 (35 S) $Shh^{-/-}$ limbs but not in E9.25 (21-23 S) WT or posterior E10 (28-30 S) $Shh^{-/-}$ limbs. Note posterior E10.5 $Shh^{-/-};Gli3^{-/-}$ limbs maintain accessible chromatin at these regions. **b** Violin plots of chromatin accessibility in WT and $Shh^{-/-}$ limbs. Globally the chromatin at HH-responsive GBRs is more accessible in posterior E10 $Shh^{-/-}$ limb buds compared to posterior E10 WT limbs and posterior E10.5 $Shh^{-/-}$ limbs. For WT E9.25 (21-23 S) and $Shh^{-/-}$ E10.5 (35 S), $n = 2$; for WT and $Shh^{-/-}$ E10 (28-30 S) and WT E10.5 (35 S), $n = 3$; for E10.5 (33-35 S) $Shh^{-/-};Gli3^{-/-}$, $n = 5$ biologically independent replicates. Two-sided Wilcoxon signed rank tests were performed to compare each pair of groups, multiple hypothesis testing adjusted using BH method. E9.25 < E10 WT, FDR < 2.69E-15; E9.25 < E10 $Shh^{-/-}$ FDR < 3.44E-23; E9.25 < E10.5 FDR < 5.82E-15 E10 WT < E10 $Shh^{-/-}$, FDR < 2.55E-4; E10 $Shh^{-/-}$ > E10.5 $Shh^{-/-}$, FDR < 1.21E-18; E10.5 WT > E10.5 $Shh^{-/-}$, FDR < 9.81E-12. Red dashed lines indicates E10.5 $Shh^{-/-}$ mean. **c** Average ATAC profiles of HH-responsive GBRs in posterior E10 and E10.5 $Shh^{-/-}$ limb buds. **d** Violin plots depicting log2 fold changes in chromatin accessibility (WT versus $Shh^{-/-}$) at E10 and E10.5 ($p = 0.000082$ two-sided Wilcoxon signed rank test). **e, f** Scatter plots of ATAC-seq signal in WT vs. $Shh^{-/-}$ limbs at E10 **e** and E10.5 **f**. GBRs and their associated genes annotated in red signify a significant reduction of chromatin accessibility in $Shh^{-/-}$ limbs compared to WT (FDR < 0.05). GBRs in blue indicate visibly reduced accessibility in E10 $Shh^{-/-}$ limb buds that are not significantly changed until E10.5 **g** ATAC-seq tracks of GBRs near $Ptch1$ that are reduced, but not significant, in E10 $Shh^{-/-}$ limbs compared to E10 WT limb buds. **h, i** GBRs that are already have significantly reduced accessibility in E10 $Shh^{-/-}$ limbs compared to E10 WT. **j** Schematic showing the temporal onset of chromatin compaction at GBRs in relation to the initiation of HH signaling. **k** Quantification of GLI3 present along ciliary axonemes out of total number of cilia (marked by ARL13b) that colocalize with GLI3 in E9.25 (21-23 S) ($n = 3$), E9.75 (26-28 S) center limb buds ($n = 6$) and E10.5 (35 S) anterior and posterior limbs ($n = 3$). Error bars indicate SEM. Unpaired, two-sided t-tests were performed. E9.25 (21-23 S) vs E9.75 (26-28 S), $p = 0.0043$; E9.75 (26-28 S) vs E10.5 (35 S) anterior, $p = 0.00025$; E9.25 (21-23 S) vs E10.5 (35 S) anterior, $p = 0.048$; E9.25 (21-23 S) vs E10.5 (35 S) posterior, $p = 0.134$; E9.75 (26-28 S) vs E10.5 (35 S) posterior, $p = 0.00061$. **l, m** Representative images of GLI3$^{FLAG}$ ciliary distribution in E9.25 **l** and E9.75 **m** limb buds acquired on a Zeiss LSM 710 Confocal visualized using maximal intensity projections with pseudo-coloring for merged images. For E9.25, $n = 3$ and for E9.75 $n = 6$ biological replicates on individual embryos. For each replicate, ~75 cilia were analyzed and quantified. **n** Schematic depicting the temporal shift in GLI3 ciliary localization. Orange shading in all tracks indicates HH-responsive GBRs defined in Supplementary Fig. 1e. Scale bars for ATAC-seq tracks = 1 kb. Scale bars for panel **l** = 0.5 μm. Source data are provided as a Source Data file.

after the onset of HH signaling, but prior to GLI repression being fully established at E9.75 (26–28 S). Here, GLI3 ciliary localization remained at comparable levels to E9.25 limbs, as well as in anterior and posterior E10.5 (32–35 S) limbs, after GLI repression is established (Supplementary Fig. 5f). Interestingly, E9.75 (26–28 S) limb buds had a significant redistribution of GLI3 within the cilia. While at E9.25 (21–23 S) and E10.5 (32–25 S), most GLI3 was localized to the ciliary tip alone with ~20–40% having signal along the ciliary axoneme, at E9.75 (26–28 S), 74% of cilia had GLI3 signal along the ciliary axoneme with a corresponding reduction in the percentage of cilia with tip restricted GLI3 (Fig. 5k, l; Supplementary Fig. 5g). This increase in ciliary axoneme distribution is suggestive of an increase in the rate of GLI ciliary trafficking, which occurs at a time just prior to when we observe initial GLI-dependent chromatin compaction and the onset of GLI repression. Overall, this data is consistent with the possibility that there may be a 'steady state' of low to moderate levels of GLI3 ciliary trafficking, with a spike in the rate of trafficking coinciding with the requirement for GLI3 repression to become established during limb development shortly after the induction of HH signaling.

**GLI3 is unlikely to co-regulate HAND2 targets in early limbs.** $Hand2$ is one of the best characterized GLI3 repression targets in the limb bud, however in opposition to this, HAND2 and GLI3 have recently been shown to physically interact and synergistically activate targets in developing facial structures[45]. Similar to their co-occurrence in craniofacial tissue[45], they are co-expressed throughout the pre-HH limb, prior to HAND2 and GLI3 segregating into distinct domains post-HH signaling[14]. Instead of being responsible for repressing targets prior to HH signaling, we questioned whether GLI3 might initially co-activate targets with HAND2, in a manner similar to that in facial tissue. Interestingly, while GLI3 bound to HAND2 binding regions both pre- and post-HH signaling near many genes (Supplementary Fig. 6a), other genes, including posteriorly expressed $Tbx2$ and $Tbx3$, were specifically bound by GLI3 at HAND2 binding regions in pre-HH but not post-HH limb buds (Supplementary Fig. 6b, light blue shading, Supplementary Fig. 8b). To determine whether GLI3 and HAND2 might initially co-activate targets specifically in the early limb, we identified 310 previously

characterized HAND2 limb binding sites[14] that are bound by GLI3 pre-HH at E9.25, but not at E10.5 (Supplementary Fig. 6b, c). We analyzed published HAND2 and GLI3 motifs[10,45] in ATAC-seq footprints to identify the co-occurrence of these factors in the early limb compared to post-HH $Shh^{-/-}$ limbs, which have active GLI repression and would not be expected to have HAND2 and GLI3 co-regulation. Inconsistent with E9.25 specific co-regulation of HAND2 and GLI3, there was no difference in HAND2 motif enrichment in ATAC footprints between pre-HH WT and post-HH $Shh^{-/-}$ (Supplementary Fig. 6c–e, Fisher's extract test $p$-value = 0.14). It is possible that other factors could be present at some ATAC-seq footprints in $Shh^{-/-}$ limbs overlapping HAND2 motif regions, thus masking our ability to detect a loss of HAND2 footprints. We also note that $Tbx2$ and $Tbx3$ are not reduced in pre-HH $Gli3^{-/-}$ limb buds as would be expected if GLI3 had a significant role in co-activating these HAND2 target genes (Supplementary Data 3). While we cannot rule out the possibility of GLI3-HAND2 co-regulation at specific enhancers, we do not believe this to be a broad mechanism for initial regulation of GLI target genes.

## Discussion

GLI repressors have been proposed to play a significant role in 'pre-patterning' the anterior-posterior limb bud prior to the onset of HH signaling[8,9,14,16,17,29,46,47]. Unexpectedly, we find that GLI3 does not act as a transcriptional repressor in the early limb bud before the onset of HH signaling. Although the GLI3-R isoform is produced at comparable levels and binds to chromatin, it does not mediate deacetylation of H3K27ac or chromatin compaction at enhancers, as it does after HH signaling[10]. Moreover, there is little to no upregulated gene expression in early $Gli3^{-/-}$ limb buds. The lack of GLI repressor activity prior to the onset of HH signaling is inconsistent with the current pre-patterning model, instead it suggests that the repressive patterning activities of GLI3 in the limb bud are initiated concurrently with, or after, the initiation of HH signaling (Fig. 6). More broadly, these findings indicate that GLI repression is not a default state prior to the onset of HH signaling and that its activation is instead likely to be context dependent (Fig. 6c).

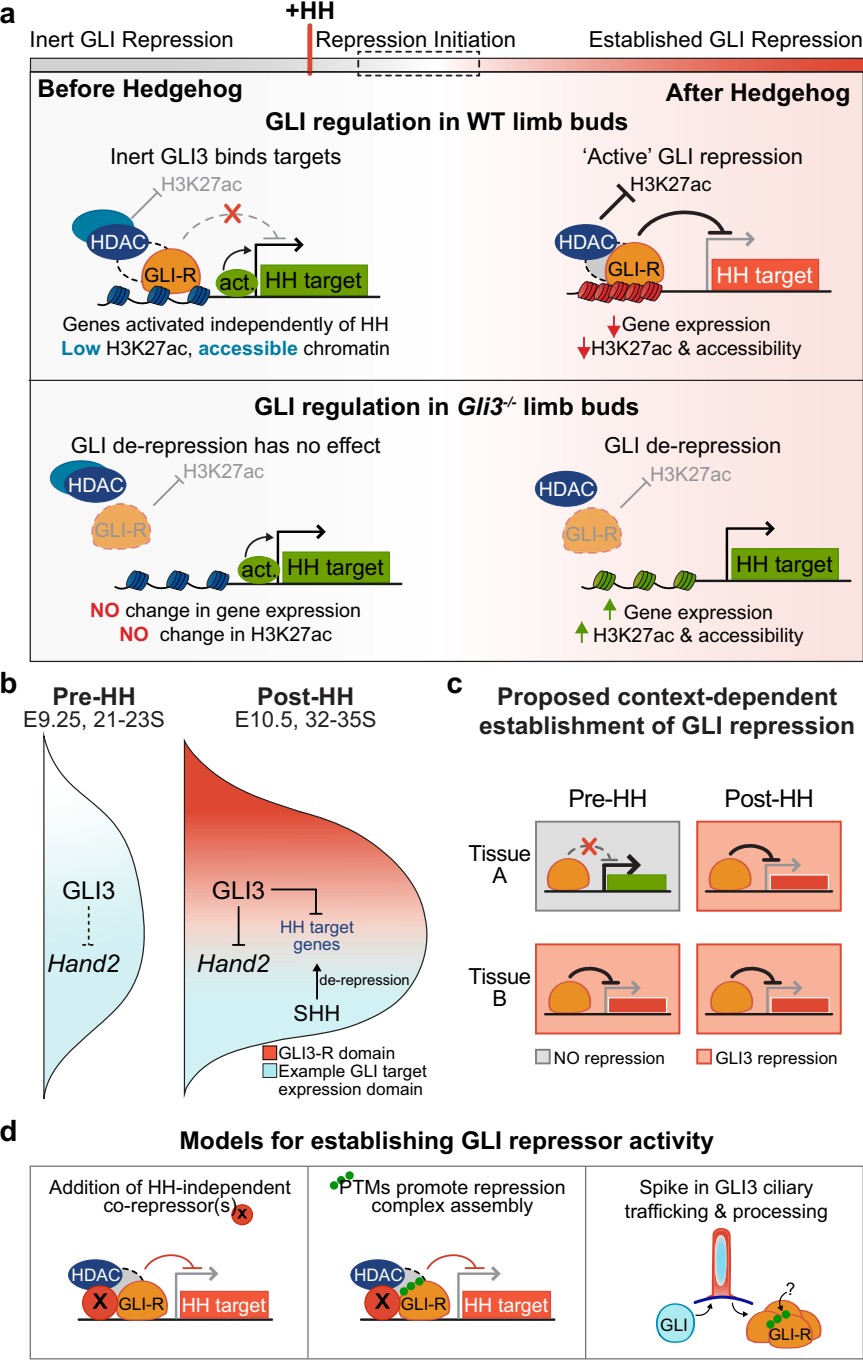

**Fig. 6 Model for establishing GLI3 repression. a** Model for lack of GLI3 repression prior to HH signaling. In pre-HH limb buds, many target genes are already activated by HH-independent factors. While GLI3 binds to most targets, it is inert as it is unable to regulate deposition of H3K27ac at enhancers or chromatin accessibility. After HH induction when 'active' GLI3 repression has been established, GLI3 prevents addition of H3K27ac at enhancers to spatially restrict expression of its target genes and pattern the limb. **b** Schematic of GLI3 spatial regulation of target genes. In pre-HH limb buds, GLI3 does not repress targets such as *Hand2*, which is expressed in an expanded anterior domain. In contrast, GLI3 restricts expression of targets to the posterior limb bud where HH signaling is active in post-HH limbs. **c** Schematic for GLI repression being established in a context-dependent manner. **d** Possible models for initiating GLI3 repressor activity. Left: HH-independent co-repressor(s), which may not be abundant in early limb development collaborate with GLI3 to assemble a complete repression complex. Middle: Addition of post-translational modifications (PTMs), potentially added via ciliary trafficking and processing, promote assembly of, or stabilize, a GLI repression complex. Right: A spike in ciliary GLI3 processing, as reflected by increased axonemal colocalization, increases the amount of viable GLI3 repressor, potentially through addition of PTMs as suggested above.

**GLI3-R is transcriptionally inert in early limb buds**. The absence of GLI repression in the early limb bud could be caused by inert GLI-R isoforms or by target genes that are not yet competent for GLI repression (Fig. 4i, j). Our results are consistent with the co-occurrence of both of these mechanisms. GLI3

does not bind to almost a third of all HH-responsive GBRs that will be bound at E10.5, and most unbound regions having not yet acquired a poised conformation (Fig. 1; Supplementary Fig. 1g, h; Supplementary Fig. 7a, b). In this case, pioneer factors may be required to prime this subset of GBRs before GLI3 can regulate

them, as has been proposed for SOX2's role in activating GLI neural enhancers[48,49]. Although GLI3 itself could act as a pioneer factor, we do not favor this possibility. First, while many regions that lack GLI3 binding also lack poised enhancer modifications, some regions remain accessible and poised, but are not yet bound by GLI3 at E9.25 (Fig. 1f–i). This suggests that GLI3 binds preferentially to enhancers that are already poised, rather than playing a role in promoting deposition of these poised modifications. Additionally, while chromatin compaction of GBRs is GLI3-dependent, initially gaining accessibility is not, as we find that in E10.5 $Shh^{-/-};Gli3^{-/-}$ limbs, the chromatin is accessible, similar to E10.5 posterior WT controls (Fig. 5).

The presence of the polycomb repressive mark, H3K27me3, at the promoters of many predicted HH target genes, provides an additional mechanism of GLI3-independent repression, as it is highly enriched at many promoters in E9.25 limbs, but is greatly reduced or absent from post-HH limb buds (Fig. 4f–j; Supplementary Fig. 4). Notably, GLI3 de-repression is unlikely to mediate this reduction, as many of these genes also lack H3K27me3 in E10.5 $Shh^{-/-}$ limb buds[10]. This suggests that removal of H3K27me3 is largely independent of HH signaling and GLI3 regulation, despite reports demonstrating that H3K27me3 enrichment at some HH targets is resolved in a HH-dependent manner (Fig. 4f–j, Supplementary Fig. 4)[50]. The H3K27me3 clearing at these genes is most likely mediated by the loss of unknown, GLI-independent transcriptional repressors as has been proposed within the developing brain[51].

Unlike the previous examples, the class of genes that are bound by GLI3 and not enriched for H3K27me3 are more likely to be competent for GLI repression in pre-HH limb buds (Fig. 4j). However, these genes are also not upregulated in $Gli3^{-/-}$ limb buds (Fig. 4b, d), suggesting an absence of GLI3 repressor function at this stage. Several additional lines of evidence support this suggestion. First, $Hand2$ is broadly expressed in the anterior domain of pre-HH limb buds where there is co-localization with GLI3-R that persists through E9.5[14] (Fig. 4d, e). Second, H3K27ac levels at HH-responsive GBRs do not increase in pre-HH $Gli3^{-/-}$ limb buds (Fig. 3) as they do in post-HH $Gli3^{-/-}$ limb buds, even at genes that appear competent for GLI-repression[10] (Figs. 3b, e; 4j). Third, while most HH-responsive enhancers have lower H3K27ac enrichment in pre-HH limb buds than in post-HH limb buds (Fig. 2b, c), exceptional enhancers with higher levels of H3K27ac do not further increase acetylation levels in $Gli3^{-/-}$ limb buds, excluding the possibility that lack of increased H3K27ac levels are due to lack of a HH-independent activator (Supplementary Fig. 2b–d). Finally, HH-responsive GBRs are broadly accessible at pre-HH stages and up until ~10 h after the time of HH induction, in contrast to the loss of accessibility seen under conditions of maximal GLI repression in post-HH E10.5 limbs (Fig. 5).

We predict that HH-independent mechanisms may be required to activate them prior to GLI3 repression becoming established. Most genes that are regulated by GLI3 in the post-HH limb are already expressed in E9.25 limb buds (Supplementary Fig. 3d), suggesting that genes might have to become activated prior to GLI3 repressing them.

**Mechanisms for establishing GLI-repressor activity**. One mechanism for regulating repressor activity is the variable expression of required co-factors as exemplified by the Brinker repressor[52]. Consistent with this scenario, the genes encoding several co-repressors implicated with GLI repression in various contexts, including $Ski$, $Smarcc1$ and $Atrophin1$[53–55] are significantly downregulated prior to HH induction compared to at E10.5 (Supplementary Data 3). Conversely, a protein enriched in

early limb buds could inhibit GLI3 repressor activity, as has been shown for HOXD12 in post-HH limb buds[56]. Although HDACs are enriched at GBRs in both pre- and post-HH limb buds (Fig. 2e, g), the lack of increased H3K27ac enrichment in $Gli3^{-/-}$ limb buds suggests that they are either inactive or not regulated by GLI3. Interestingly, both HDAC1 and HDAC2 initially co-localize at many GBRs at E9.25, while in post-HH limb buds, most remain bound by HDAC1 but lose HDAC2 enrichment. In other contexts, HDAC2 is required to recruit, but not maintain HDAC1 at some enhancers[57], and a similar mechanism could enable the assembly of an HDAC1-containing GLI repression complex. As HDACs require co-repressor complexes to guide them to their substrates[58], the absence of a functional GLI3 co-repressor might prevent HDACs from properly regulating GLI3 enhancers in the early limb. Alternatively, rather than missing a co-repressor, it is possible that GLI3-R lacks additional, uncharacterized post-translational modifications required for its activity, or to functionally interact with its repression complex (Fig. 6d).

**Temporal onset of GLI transcriptional repression**. GLI-dependent restriction of chromatin accessibility can first be detected by ATAC-seq at 28–30 S when a few HH-responsive GBRs have significantly reduced accessibility in $Shh^{-/-}$ limb buds. However, most regions are actually more accessible than in E10.5 $Shh^{-/-}$, suggesting that GLI-mediated compaction is not widespread at this time (Fig. 5). Focusing on $Hand2$ as a likely direct target of GLI3 repression[9,14,15], there is no reduction in chromatin accessibility at this locus in E10 $Shh^{-/-}$ limbs as is seen at E10.5 (Fig. 5a). However, GLI3 transcriptional repression of $Hand2$ is likely to commence around 26 S (Fig. 4e)[14], a time-point coinciding with the activation of $Shh$ expression in the limb bud (Fig. 1a, b; Supplementary Fig. 1a, b)[18,20]. The lack of detectable reductions in chromatin accessibility at this time indicate that GLI-dependent compaction likely occurs as a later step in GLI3-mediated repression.

The onset of GLI transcriptional repression is accompanied by a re-distribution of ciliary GLI3 into the axoneme, suggestive of a possible increase in GLI3 trafficking that coincides with the time when GLI3 starts to functionally repress target genes. We propose that increased GLI3 trafficking is facilitated by developmental changes to unknown ciliary processing components that regulate GLI3-R activity (Fig. 6d). Since there are comparable levels of GLI3-R protein present within the pre- and post-HH limb buds (Fig. 1c), this regulation seems most likely to affect an unknown post-translational modification of GLI3-R rather than the processing of GLI3-FL into truncated GLI3-R (Fig. 6d). If this is the case, then the activation of GLI repression is dependent on the status of the cilia and GLI transcriptional repression may be dynamically regulated in different developmental contexts. Consistent with this notion, GPR161 has recently been shown to repress HH signaling by regulation of GLI-R processing, through both ciliary and extraciliary mechanisms in a tissue-specific manner[59].

**GLI2 is unlikely to serve as a repressor in the absence of GLI3**. One possible explanation for the lack of GLI3 repressor activity is functional redundancy with other GLI proteins. While $Gli1$ is not expressed in pre-HH limb buds (Fig. 1b, Supplementary Fig. 1a, b), GLI2 is present (Supplementary Fig. 5c, e). GLI2 acts as a transcriptional activator in most embryonic tissues[4,60–63] but in certain contexts has weak repressor properties[4,60,64–66]. Loss of both GLI2/3 in $Gli2^{-/-};Gli3^{-/-}$ limbs do not further alter digit patterning in $Gli3^{-/-}$ limbs[67], suggesting that GLI2 repression does not regulate limb patterning. Similarly, there is no evidence for GLI2-R compensating for GLI3-R in $Shh;Gli3$ double mutants,

either on the gene or enhancer level post-HH signaling[7–10]. Overall, this indicates that GLI2-R does not regulate post-HH limb patterning though it does not rule out a scenario where transient GLI2-R might initially be redundant with GLI3-R in pre-HH limb buds. However, we think this is unlikely for two reasons. First, in pre-HH WT limb buds, the GLI3-R target *Hand2* is initially co-expressed with *Gli3* throughout the entire limb bud before becoming posteriorly restricted by GLI3 in later limb development[14] (Fig. 4e; Supplementary Fig. 3e). In these WT pre-HH limbs, GLI2-R does not function to prematurely repress *Hand2*, one of the signature repression targets of GLI3. Second, GLI3-dependent reduction in chromatin accessibility at HH-responsive GBRs is a hallmark of GLI repression[10] (Fig. 5a). However, there is no chromatin compaction in pre-HH WT limb buds or in *Shh*$^{-/-}$ limb buds (with constitutive GLI2 and GLI3 repression), until nearly 12 h after HH signaling would have normally been activated (E10.0, 28–30 S) (Fig. 5a–j).

**A model for establishing GLI3-mediated repression**. Our findings provide genetic and genomic evidence that GLI3-R is inert prior to HH induction in the limb bud. As GLI3-dependent chromatin compaction is established in E10.5 (35 S), but not E10 (28–30 S) *Shh*$^{-/-}$ limb buds, it also suggests that GLI repression is likely established temporally during limb development through HH-independent mechanisms. Rather than preventing initial expression of HH targets, most genes repressed by GLI3 in later limb development are initially expressed in pre-HH limb buds (Supplementary Fig. 3d; Supplementary Data 3). Once repression is later established through unknown mechanisms, GLI3 functions to restrict target gene expression to the posterior limb (Fig. 6b, d). More broadly, this work indicates that GLI repression is not a default state and instead must be acquired in a context-dependent fashion. Since we have demonstrated that GLI repression is established independently of HH, tissue-specific factors may play a role in initiating GLI repression. Thus, repression may be established at different times, in relation to HH activation, depending on the tissue type (Fig. 6c). This work has major implications for our understanding of the pathway itself, specifically how GLI repression might be regulated in context-dependent mechanisms and whether genes and enhancers must gain competency to be regulated by HH signaling.

## Methods

**Mouse Strains and Embryonic manipulations**. Experiments involving mice were approved by the Institutional Animal Care and Use Committee at the University of Texas at Austin (protocol AUP-2019-00233). The *Gli3*$^{Xt-J}$ (Jackson Cat# 000026) and *Shh*$^{tm1amc}$ null (Jackson Cat# 003318) alleles[68,69] were maintained on a Swiss Webster background. The *Gli3*$^{3XFLAG}$ allele, with an N-terminal 3XFLAG-epitope[26,70], was maintained on a mixed background. Mice were housed in a facility with 12 h light/ 12 h dark cycles and temperatures of 65-75 °F (~18–23 °C) with 40–60% humidity. For all genomic experiments, with the exception of *Shh*$^{-/-}$;*Gli3*$^{-/-}$ ATAC samples which had to be individually genotypes, fresh E9.25 (21–23 S) and E10.5 (32–35 S) forelimb buds were pooled from multiple litters of the specified genotype. For ATAC-seq experiments with WT forelimbs (28–30 S; 35 S), and *Shh*$^{-/-}$ forelimbs (28-30 S; 35 S), posterior halves of forelimbs from 2 to 3 embryos were dissected and pooled prior to cell dissociation. For RNA-seq, E9.25 whole forelimbs or E10.5 anterior halves of WT and *Gli3*$^{-/-}$ forelimbs were pooled from 3 to 4 embryos and dissociated in TRIzol (Invitrogen Cat# 15596026). Purified RNA was library prepped using NEB-Next Ultra II RNA library Prep Kit for Illumina (NEB Cat# E7775) and sequenced with an average of 60 million reads/sample; 3–4 biological replicates were used for each stage and genotype.

**In situ hybridization**. Conventional in situ hybridization done as previously described[7]. Briefly embryos were fixed for 16 h at 4 °C in 4%PFA/PBS and rehydrated through methanol/NaCl washes. Embryos were treated with proteinase K (10 μg/ml) for ~15 min, post-fixed, and prehybridized at 70 °C for 1 h before being hybridized with 500 ng/ml digoxigenin-labeled riboprobes at 70 °C overnight. Samples were then incubated with RNase A, washed, blocked and incubated with a 1:5000 dilution of anti-digoxigenin antibody conjugated with alkaline phosphatase (AP) (Roche) overnight at 4 °C. Extensive washing in 1XMBST was followed by

incubation in NTMT prior to signal visualized with BM purple. Fluorescent in situ hybridization was performed using HCR v3.0 Molecular Instruments reagents and was performed according to manufacturer's protocol[71] except that probes were incubated overnight with a concentration of 16 nM. Embryos were embedded in Ultralow gelling temperature agarose (Sigma A5030), cleared in Ce3D + +[72] and visualized on a Zeiss LSM 710 confocal microscope and are shown as maximal intensity z-stack projections.

**Immunofluorescence**. For M2 FLAG ciliary staining, freshly dissected, unfixed embryos were sucrose protected (30% sucrose) at RT and embedded in OCT. 12 μm cryosections were immediately fixed in 100% methanol for 3 min at −18 °C. For GLI2 ciliary staining, embryos were fixed with 4% PFA/PBS at RT for 1 h, prior to sucrose protection and embedding. Sections were permeabilized in 0.1% PBST at RT for 15 min. then blocked in 3% BSA (Fisher Scientific; BP1600-100), 5% Normal Goat Serum (Jackson Immunoresearch #005-000-121) in 0.1% PBST for 1 h. at RT. Primary antibodies were incubated in blocking solution overnight at 4 °C at 1:500 dilutions: M2 FLAG (Sigma-Aldrich; F3165), Arl13b (Proteintech; 17711-1-AP), GLI2 (a gift from Jonathon Eggenschwiler), Gamma tubulin (Sigma #T6557). Secondary antibodies were incubated in block at 1:500 dilutions at RT for 1 h: goat anti-mouse Alexa568 (Thermo Fisher Scientific; A-11004), goat anti-rabbit Alexa488 (Thermo Fisher Scientific; A-11034) and donkey anti-guinea pig Alexa594 (Jackson ImmunoResearch 706-585-148). After washes, sections were then stained with 1:5,000 DAPI in PBS for 8 min at room temperature prior to being coverslip mounted with Prolong Gold mounting media (Invitrogen #P36930). Images were taken with a Zeiss LSM 710 Laser Confocal at 63x. Images were quantified using Zen Blue Lite software to notate category designations. Cilium length measurements were made in FIJI (Image J) normalized to a Zeiss scale bar graphic.

**Western Blots**. Western blots were blocked in 5% milk for 30 min., then incubated with primary antibodies for 1 h at room temperature in 3% milk: 1:4000 M2 Flag (Sigma #F3165) and 1:2000 B-actin (Cell Signaling #8457). Secondary antibodies were incubated for 1 h at room temperature in 3% milk: 1:5000 Donkey anti-mouse (Jackson #715-035-150), Donkey anti-rabbit (Jackson #711-005-0152). Blots were developed using Amersham ECL Prime (GE Healthcare #GERPN2232).

**Chromatin Immunoprecipitation**. ChIP-Seq was performed on whole E9.25 (21-23 S) forelimbs pooled from 30-40 embryos as previously described[15] with the following modifications. Cells were dissociated with 100μg/ml Liberase (Roche 05401119001) and fixed for 15 min in 1% formaldehyde. After cell lysis, chromatin was sheared in buffer containing 0.25% SDS with a Covaris S2 focused ultra-sonicator. Antibodies used for ChIP experiments include H3K27ac (3μg/reaction; Abcam #ab4729) and H3K4me2 (3μg/reaction; Millipore #07-030). Libraries were generated using the NEBNext Ultra II library preparation kit with 15 cycles of PCR amplification (NEB E7645) and sequenced to a depth of >40 million reads per sample, using two biological replicates. Peaks were called using CisGenome version 2.1.0[73]. The read numbers were adjusted by library size and log2 transformed after adding a pseudo-count of 1. The differential analyses were performed using Limma version 3.14[74]; peaks with an FDR < 0.05 were considered to have differential enrichment.

**ATAC-Seq**. WT E9.25 (21-23 S) forelimbs or posterior forelimbs from E10 (28–30 S) or E10.5 (35 S) forelimbs were dissected and pooled from WT embryos ($n = 3$, $n = 3$). Posterior forelimbs from E10 (28–30 S) or E10.5 (35 S) were pooled from *Shh*$^{-/-}$ embryos ($n = 3$, $n = 3$). ATAC used components from the Nextera DNA Library Preparation Kit (Illumina) as described previously[75] with the following variations. Limb buds were dissociated using 100μg/ml Liberase (Roche 05401119001) and lysed for 10 min. at 4 °C, rotating, in a nuclear permeabilization buffer (5% BSA, 0.2% NP-40, 1 mM DTT in Ca + /Mg+ free PBS + protease inhibitor) and then centrifuged for 10 min. at 300 g, 4 C. Nuclei were resuspended, counted using Trypan Blue, and 30,000 nuclei from each WT and *Shh*$^{-/-}$ sample were put into each transposase reaction. The transposase reaction (30 μL volume) was carried out for 1 h at 37 °C with gentle agitation. Libraries were generated using 11 cycles of PCR amplification with NEB high fidelity 2x master mix (New England Biolabs), cleaned up with SparQ PureMag beads (QuantaBio) and sequenced on an Illumina NovaSeq using PE150 or SR100 to a depth of at least 40 million reads. Peaks were called using MACS2 with a fixed window size of 200 bp and a q-value cutoff of 0.05. Read counts from the peak regions were obtained and differential analysis was performed using DESeq2[76]. Peaks with FDR < 0.05 in the differential test were considered to have significant chromatin accessibility difference.

Footprinting analysis was performed using HINT[77] based on the ATAC-seq data. After obtaining the DNA footprints, GLI3 and HAND2 motifs were mapped to the footprints which overlap with regions that have pre-HH only GLI3 binding, E10.5 HAND2 binding, and E9.25 H3K27ac signals (Supplementary Fig. 6C) using CisGenome version 2.1.0[73]. Motif enrichment (Supplementary Fig. 6E) is calculated as the ratio between the percentage of footprints containing the motif in the E9.25 WT samples and that in the E10.5 *Shh*$^{-/-}$ samples. Fisher's extract test is applied to test if the motif enrichment is different between E9.25 WT and E10.5 *Shh*$^{-/-}$ (GLI3 *p*-value = 0.87, HAND2 *p*-value = 0.14).

**CUT&Tag**. Experiments were performed as described previously[39] with the following modifications. 100,000 cells from E9.25 (21–23 S) forelimbs were dissociated, bound to Concanavalin beads and incubated overnight at 4 °C on a nutator with primary antibodies. Antibodies were used at the following concentrations: 1 μg/mL of H3K4me1 (Abcam #ab8895) and 5 μg/mL of H3K27me3 (abcam #ab195477). The following day, samples were incubated at room temperature for 30 min. with secondary antibody, 1:100 Donkey anti-rabbit (Jackson ImmunoResearch #711-005-152). CUT&Tag transposases used were gifted from the Henikoff lab and EpiCyphr (now commercially available). Libraries were generated using NEB high fidelity 2x master mix with 14 PCR cycles and cleaned up to remove adapters using SparQ PureMag beads (QuantaBio). Samples were sequenced on an Illumina NextSeq 500 instrument using PEx75 to a depth of depth of 3–5 million reads. Peaks were called using SEACR[78].

**CUT&RUN**. Experiments were performed according to EpiCypher's CUTANA CUT&RUN protocol (EpiCypher #15-1016) with the following modifications. 200,000–300,000 cells from E9.25 (21–23 S) forelimbs and anterior E10.5 (32–35 S) forelimbs were incubated overnight at 4 °C on a nutator with 1:250 FLAG primary antibody (Sigma #F3165), 1:100 HDAC1 (Abcam ab7028) or 1:200 HDAC2 (Abcam ab7029). The following day, samples were incubated at room temperature for 30 min. with secondary antibody, 1:100 Donkey anti-mouse (Jackson ImmunoResearch #715-035-150) or Donkey anti-rabbit (Jackson ImmunoResearch #711-005-152), followed by three washes in Digitonin wash buffer. CUTANA pAG-MNase was then incubated with samples for 10 min at room temperature and then the MNase reaction was performed for 2 h at 4 °C on a nutator. Libraries were generated using NEBNext Ultra II DNA Library Prep Kit with 14 PCR cycles and cleaned up to remove adapters using AMPure XP beads (Beckman Coulter) or SparQ PureMag beads (QuantaBio). Samples were sequenced on an Illumina NextSeq 500 instrument using PEx75 to a depth of depth of 3–5 million reads. Peaks were called using MACS[79].

**RNA-seq**. Single limb pairs were collected, saved in TRIzol and later pooled (~4 limb bud pairs/replicate) for each developmental stage, E9.25 (21–23 s) and E10.5 (33–35 S). After RNA extraction, libraries were prepped with NEBNext Ultra II RNA Library Prep Kit. Samples were sequenced on a NovaSeq S1, SR100, with a minimum of 40 million reads/sample. Reads were aligned with HISAT2 and DESeq2 was used for DEG analysis.

**Intron retention rates**. The formula (1-N)/M was used to calculate the intron retention rate for each gene, where $N$ = the number of reads overlapping with any of the gene's exons and M = the number of reads overlapping with the body of the gene. Limma version 3.14[74] was used to identify differential intron retention rate between E9.25 and E10.5 $Gli3^{-/-}$ limbs, FDR cutoff< 0.05.

**Identification of GLI target genes**. To identify GLI target genes, we developed a weighted system which incorporated HH-responsive GBRs, poised chromatin modifications, active promoter marks and genes identified as being significantly downregulated in E10.5 $Shh^{-/-}$ RNA-seq compared to WT limb buds[7] (FDR = 0.05; WT/$Shh^{-/-}$ FC < 0) (GSE58222 [https://www.ncbi.nlm.nih.gov/geo/query/acc.cgi?acc=GSE58645]). HH-responsive GBRs were intersected with poised enhancer modifications from H3K4me1 ChIP-seq (GSE86690), H3K4me2 ChIP-seq and E10.5 ATAC-seq datasets[10] (GSE108880) from E10.5 WT limb buds. HH-responsive GBRs enriched for poised enhancers modifications, and thus increasing the probaly of being active enhancers, were given a higher weight than ones without modifications. Genes identified as downregulated in $Shh$ mutant limbs were intersected with H3K4me3 (GSE86698) E10.5 WT limb datasets. Genes enriched for these active promoter marks were given more weight. The weighted HH-responsive GBRs and weighted genes were then intersected based on proximity, up to 500 kb, and within the same topologically associated domain (TAD) (GSE96107). Genes were identified as a putative HH targets based on their proximity to HH-responsive GBRs and the number of HH-responsive GBRs, such that a gene with multiple HH-responsive GBRs, enriched for poised enhancer marks, would rank higher than a gene with HH-responsive GBRs found more distally. [https://github.com/Boksunni/Predicted-HH-List-Generation] [https://zenodo.org/badge/latestdoi/379668324].

**Quantitative qPCR**. Individual pairs of 21–35 S limb buds were dissected and dissolved in Trizol. RNA was extracted from the samples using SuperScript IV VILO Master Mix with ezDNase Enzyme (Invitrogen #11766050). Primer sequences used:
*Gapdh* F: GGTGAAGGTCGGTGTGAACG R: CTCGCTCCTGGAAGATGGTG
*Gli1* F: CCCAGCTCGCTCCGCAAACA R: CTGCTGCGGGCATGGCACTCT
*Ptch1* F: GACCGGCCTTGCCTCAACCC R: CAGGGCGTGAGCGCTGACAA

**Reporting summary**. Further information on research design is available in the Nature Research Reporting Summary linked to this article.

## Data availability

The genomic datasets generated in this study, including CUT&RUN, CUT&Tag, ATAC-Seq, ChIP-seq and RNA-seq datasets, have been deposited in the GEO under accession code GSE178838. Additional processed, including called peaks or differential analyses for these datasets are available in the Source Data. Additional datasets used in this study, include: E10.5 WT vs. $Shh^{-/-}$ H3K27ac ChIP-seq, E10.5 $Shh^{-/-}$ vs. $Shh^{-/-};Gli3^{-/-}$ microChIP-seq, E10.5 WT H3K4me2 (GSE108880) and E10.5 WT H3K4me1 ChIP-seq (GSE86690). Source data are provided with this paper.

## Code availability

The pipeline used to identify putative direct HH target genes can be found at [https://github.com/Boksunni/Predicted-HH-List-Generation] [https://zenodo.org/badge/latestdoi/379668324].

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

## Acknowledgements

We thank Matt Anderson and Mark Lewandoski for sharing their protocol for clearing embryos, We thank the Henikoff Lab and EpiCypher for providing CUT&Tag reagents, and Jonathan Eggenschwiler for providing the GLI2 antibody. We thank Jessica Podnar from the Genomic Sequencing and Analysis Facility at the University of Texas at Austin,

Anna Webb from the Center for Biomedical Research Support at the University of Texas at Austin and Elle Roberson for technical advice, and Janani Ramachandran for comments on this manuscript. This work was supported by NIH R01HD073151 (to S.A.V. and H.J.), F31DE027597 (to R.K.L.) and R01HG009518 (to H.J.).

## Author contributions

Conceptualization, R.K.L. and S.A.V.; Methodology, R.K.L, W.Z., Z.J., Software J.D.K.; Validation K.E.W.; Formal Analysis, R.K.L., W.Z., Z.J.; Investigation, R.K.L., W.Z., Z.J., K.N.F., K.E.S., K.E.W.; Data curation, R.K.L., W.Z.; Writing—Original Draft, R.K.L. and S.A.V.; Writing—Review & Editing, S.A.V, R.K.L., W.Z., H.J.; Supervision, S.A.V., H.J.; Funding Acquisition, S.A.V., H.J., R.K.L.

## Competing interests

The authors declare no competing interests.
