## [Peer Review File · Nature Communications]

GLI transcriptional repression is inert prior to Hedgehog pathway activationReviewers' Comments:

Reviewer #1:

Remarks to the Author:

In this paper the authors search to understand how Gli repression is established during limb development as whether it plays a significant role in limb pre-patterning, before Shh activation. To this end, the authors start by precisely defining the time of Shh activation, as per Gli1 expression, and establish two distinct time points, pre-HH (21-23 So, E9.25) and post-HH (E10.5) for further study. They find that in the pre-hh limb stage Gli3 binds to the majority (82%) of all GBS but only to 70% of the 309 hh-responsive genes they previously identified. Most of the Gli3 bound hh-resp GBR correspond to poised regions defined as accessible and enriched for H3K4me1 and/or H3K4me2. Acetylation is reduced at hh-responsive GBRs pre-HH but unexpectedly it does not increase upon Gli3 removal. Also, Gli3 target genes are not activated with loss of Gli3, strongly indicating that Gli3 repression is not occurring in the early limb bud. Considering a subset of Gli target genes, the authors find H3K27me3 enrichment in the promoters of most of them showing PCR2 repression independent of Gli regulation. Gli3 removal do not result in gain of acetylation, even in those target genes with no H3K27me3 enrichment, leading to the conclusion that Gli3 repressor is inert in the early limb bud. Also, the examination of an intermediate stage (E10), shows that the onset of Gli3 mediated chromatin compaction and reduced accessibility is not significant until E10.5.

My overall opinion is that this is a well conducted and very comprehensive study that provides compelling evidence of Gli3 not acting as a transcriptional repressor in the early limb bud. Therefore, I warmly recommend it for publication. The relevant genomic data presented here, and the conclusions reached, which I think are well sustained, question broadly accepted paradigms in limb development such as the early Gli3-repression of Shh targets including the Gli3 repression of Hand2 in limb pre-patterning. This study indeed opens new avenues for future studies, as considered in the discussion, including how the initial enrichment in K27 trimethylation is resolved or what is the function of Gli3 in early stages, aspects that I understand are not the scope of the current paper. I mention these specific points for consideration by the authors before publication:

- For Fig.1, I think it would be more interesting to show the initiation of expression of Gli1 and Shh (currently in Fig.S1A) than the bar graphic in Fig1B.
- The WB in Fig. 1C and Supp dataset 3 needs quantification. It seems that the level of both Gli3FL and Gli3R forms is higher in pre-HH stages. How does the Gli3 expression level in the RNA-seq datasets compare between pre and post-hh stages? It would be much informative that the analysis at E10.5 is performed separately in the anterior and posterior bud, for a better comparison, as most Gli3R is anterior.
- In the heatmap in Fig. 1G, the region marked as GBRs not bound pre-HH should correspond to 30%, however it seems smaller.
- It would be interesting to know the overlap between the GBRs not bound pre-HH (206/309) and those with called H3K27ac peaks (93/309).
- My understanding is that most of the post-HH Gli repression refers to the anterior limb, as schematized in the last figure. I think that the spatial issue (anterior vs posterior) gets hidden along the paper and would suggest the authors to clarify this point. For example, does Fig. 5B refer to the whole limb bud?
- line 38: better Gli3R?

Minor points:

- Line 121: it is not clear to what these figures refer to (86% (6385/7382)).
- Personally, I would like to see the E10.5 acetylation heatmap of Fig. 2C also included in Fig. 3B, even is repeated, for a better comparison.
- The Hoxd12 and Cdk6 GBS in Fig. 3e are not referred in the text.
- CUT&Tag for K27me3 has been performed separately in the anterior and posterior E10.5 bud. If this is the case, the specific datasets used in the figures (Fig. 4G-H) should be indicated.
- Because there is not an antibody that exclusively detects the Gli3R processed form, I would not make a strong point of the ciliary observations. Is there any concomitant change in the amount of Gli3

signal in the nucleus? Also, could the spike in the rate of ciliary Gli3 trafficking at E9.75, be detected in WBs?

- Supplementary Fig. 3D is not mentioned. The caption refers to Supplementary Fig. 3E.

Reviewer #2:

Remarks to the Author:

In this manuscript Lex and collaborators study the function of Gli3 during limb bud formation in mice. Using different genomic techniques and different mutant mouse models, the authors clearly establish that the Gli protein does not act as a repressor of its target genes before the HH signal starts to function during limb development. Thus, the authors demonstrate an "inert" role of Gli in the early stages of limb development. This inert role is given because Gli is present but its loss does not increase the expression of the target genes nor the acetylation or accessibility of their enhancers. On the other hand, repression of target gene expression by Gli is established independently of HH signaling but only once HH signaling has begun. These results are of interest both for the knowledge of mouse limb development and patterning, and for the understanding of the function of the HH-Gli signaling pathway. For these reasons I think that this study can be accepted for publication in Nat Comm.

I have a few minor comments that could improve or clarify some aspects of the manuscript.

- For readers who are not experts in the mouse model, the use of different terms to refer to developmental stages is quite confusing. Sometimes authors refer to them as E10, E10.5 etc and sometimes as a function of the number of somites. A homogenization of these terms in the manuscript could facilitate understanding. Personally, I prefer to use the number of somites as a reference, since the same stage (E10) may contain embryos with a greater or lesser number of somites.
- HH signalling activation is established by looking at Gli1 expression by in situ hybridization. This technique is not highly accurate and maybe RTqPCR could give a more precise moment for the HH activation.
- In line 75, the authors say that Gli3-R is expressed at comparable levels at pre- and post-HH stages. From the Western blot presented it seems that postHH expression is lower than pre-HH expression. Would it be possible to show a quantification of the Western blot?
- Upregulation of Foxf1 and Osr1 is explained as residual mRNAs from the lateral plate mesoderm. While this explanation seems convincing to me, would it be possible to check on scRNAseq data present in different databases to show that this genes expression is specific to the lateral plate mesoderm?
- The authors use quantitative terms when talking about in situ hybridization experiments (ex. Lines 184-186). However, ISH is not quantitative. Would it be possible to quantify expression in such cases?
- While there is no difference in Hand2 motif enrichment between pre-HH and post-HH when the authors studied the 310 Hand2 binding sites bound by Gli3 all together, would it be possible that specific sites in specific enhancers may be co-regulated?
- If Gli binding to GBRs in Pre-HH limb buds is inert, but these bound regions are enriched for poised enhancer modifications? Could we consider Gli as a pioneer factor?
- Figure 6d the legend should say "Left, middle and right" instead of top middle and bottom.

Reviewer #3:

Remarks to the Author:

Shh regulates limb patterning largely by preventing formation of Gli3 repressor (Gli3R) and enabling de-repression of target genes, and cross-repression between high anterior Gli3R and posterior Hand2 is thought to play a critical role in "pre-patterning" limb A-P polarity prior to Shh activation. In this manuscript, Lex et al. use genome-wide approaches to systematically examine the role of Gli3R in the limb prior to initiation of Shh signaling (pre-HH), identifying Gli3 binding, chromatin status and expression of Shh limb target genes in WT, Gli3 KO, and Shh KO embryos at pre-Hh stage, compared

with WT, Shh KO, or Shh/Gli3 KO (no activator, no repressor; similar to pre-Hh Gli3 KO condition) at post-Hh stage (after Shh activation). They present convincing evidence that even though a large proportion of targets are 'poised' for expression (accessible) and are Gli3-bound at pre-Hh stage, targets are surprisingly not repressed by Gli3 pre-Hh. Even Shh KO limb buds (with high Gli3R) appear not to have target repressive chromatin changes at a pre-Hh stage. In agreement, targets are not precociously de-repressed in the Gli3 KO; ectopic anterior activation in the Gli3 mutant commences post-Hh. On the other hand, a number of target genes do have PRC2 marks (H3K27me3), and/or "closed" chromatin, suggesting that other factors mediate pre-Hh repression. Focusing on pre-Hh Hand2 regulation, as a signature Gli3R target that is expressed pre-Hh as well as post-Hh, they show that Gli3 and Hand2 are extensively co-expressed pre-Hh and neither is strictly localized as proposed by the pre-patterning model. Although Gli3 is bound to Hand2 pre-Hh, it is functionally inert until the transition to post-Hh, which nevertheless occurs independent of Shh signaling (present in later stage Shh KO).

These results make a convincing case that Gli3 repression only becomes active following Shh activation and challenge the long-standing prepatterning model for initiation of limb A-P polarity. Like other paradigm-shifting discoveries, this paper raises more questions than it answers: what regulates Hh-targets prior to pathway activation, and how is the change in Gli3R function triggered during this transition. The results likewise raise the possibility that similar regulation (lack of pre-Hh "competency" for Gli3R) applies in other contexts where the Shh pathway plays a major patterning role.

This work has potentially broad ramifications and will appeal to anyone interested in Hedgehog signaling. I have some suggestions below that the authors may wish to consider for improving the clarity and readability/accessibility of the paper.

There is a great deal of (well-executed) data presented and summarized; it would be very helpful in digesting the data to know how particular 'signature' targets actually behave (which can't be inferred from Venn diagrams and heatmaps). The tabular summary of H3K27 methylation and acetylation profiles for the list of 74 high confidence targets (data set 5) is very useful in this regard; it would be even more helpful to add columns also summarizing the Gli3 binding status and chromatin accessibility status (open or closed) in the pre-Hh limb bud.

Related to the above point, for different types of analyses, peak profiles for different gene target examples are often chosen. Much of the detail understandably focuses on Hand2, which is distinctive in already being highly expressed pre-Hh. Likewise, Ptch1 and Gli1 are a special subclass that is uniquely GliA dependent. It would be nice to also show the complete set of peak profiles (Gli3, ATAC, H3K27me3, H3K27Ac, in WT and in mutants) for a couple of selected high-confidence targets that are not highly expressed pre-Hh but are Gli3R-regulated after Shh activation, and include the major different sub-classes in data set 5 (for eg. Grem1, Hoxd13, Hhip, Ets2). This would highlight the differences among subclasses more clearly. For example, Grem1 has closed chromatin early, yet surprisingly also has appreciable H3K27Ac; presumably the reverse is the case for some other targets such as Hoxd13.

The authors argue that Gli2, which can have a weak repressor activity in certain contexts, does not likely compensate for the apparent lack of Gli3R function at pre-Hh stage. Some of these arguments are more convincing than others.

--Whether the Gli2/3 double KO has no further digit patterning phenotype is debatable; at early-stage removal, digit number is certainly increased beyond Gli3 KO alone and "patterning"/morphology changes per se are more difficult to assess, considering other late (Ihh) effects.

--The argument that Gli2R function isn't evident in Shh/Gli3KO limbs (which are similar to the Gli3 KO) assumes that Gli2R function is also not evident in the Gli3 KO, but this is not really known (the Gli2/3 KO has not been analyzed in any detail in terms of early target gene expression compared to Gli3 KO).

--Finally, the authors do note that even in Gli3R-regulated targets post-Hh, compaction due to Gli3R seems to occur later, after the onset of actual repression by Gli3R. Together with the other evidence in this paper that repression and compaction are not always synonymous (PRC2+ and open chromatin on some targets pre-Hh), this raises a caveat for concluding absence of both Gli2R and Gli3R function.

In the discussion, the authors speculate on how targets may be repressed pre-Hh, which could be mediated by PRC2 for those with H3K27 methylation or bivalent marks, or may occur because pioneer factors are required to activate those that have inaccessible, closed chromatin. But a third class remains, as the authors note, that has open chromatin and is not PRC2-regulated, and yet is not highly expressed pre-Hh. How do they envision that this class is regulated?

minor comments:

Suppl Figure 3 -- the legend is missing a description for panel D, and the panel E image is described as D in the legend.

line 378-9: sentence is poorly phrased (unclear pronoun reference).

REVIEWER COMMENTS

Reviewer #1 (Remarks to the Author):

In this paper the authors search to understand how Gli repression is established during limb development as whether it plays a significant role in limb pre-patterning, before Shh activation. To this end, the authors start by precisely defining the time of Shh activation, as per Gli1 expression, and establish two distinct time points, pre-HH (21-23 So, E9.25) and post-HH (E10.5) for further study. They find that in the pre-hh limb stage Gli3 binds to the majority (82%) of all GBS but only to 70% of the 309 hh-responsive genes they previously identified. Most of the Gli3 bound hh-resp GBR correspond to poised regions defined as accessible and enriched for H3K4me1 and/or H3K4me2. Acetylation is reduced at hh-responsive GBRs pre-HH but unexpectedly it does not increase upon Gli3 removal. Also, Gli3 target genes are not activated with loss of Gli3, strongly indicating that Gli3 repression is not occurring in the early limb bud. Considering a subset of Gli target genes, the authors find H3K27me3 enrichment in the promoters of most of them showing PCR2 repression independent of Gli regulation. Gli3 removal do not result in gain of acetylation, even in those target genes with no H3K27me3 enrichment, leading to the conclusion that Gli3 repressor is inert in the early limb bud. Also, the examination of an intermediate stage (E10), shows that the onset of Gli3 mediated chromatin compaction and reduced accessibility is not significant until E10.5.

My overall opinion is that this is a well conducted and very comprehensive study that provides compelling evidence of Gli3 not acting as a transcriptional repressor in the early limb bud. Therefore, I warmly recommend it for publication.

The relevant genomic data presented here, and the conclusions reached, which I think are well sustained, question broadly accepted paradigms in limb development such as the early Gli3-repression of Shh targets including the Gli3 repression of Hand2 in limb pre-patterning. This study indeed opens new avenues for future studies, as considered in the discussion, including how the initial enrichment in K27 trimethylation is resolved or what is the function of Gli3 in early stages, aspects that I understand are not the scope of the current paper.

I mention these specific points for consideration by the authors before publication:

We thank the reviewer for their supportive evaluation of the study and comment on each of their specific points below.

- For Fig.1, I think it would be more interesting to show the initiation of expression of Gli1 and Shh (currently in Fig.S1A) than the bar graphic in Fig1B.

In response to Reviewer #2, we have now performed qPCR on individual forelimb pairs from 21-28S in addition to E10.5 WT and *Shh*^{-/-} controls, to quantify the expression of *Ptch1* and *Gli1* during the onset of HH signaling. These results are in agreement with the expression data, also showing variability in the exact time when HH signaling initiates. Since the in situs are nice visual depictions of the qPCR data, we have chosen to keep these figures together in Supplementary Fig. 1a-c so they can be directly compared. See new panel below.

Supplementary Fig. 1C. Quantitative PCR showing expression of *Ptch1* and *Gli1* in forelimb pairs from individual embryos at specified somite stage. RNA was isolated from individual forelimbs using Trizol and cDNA was generated using Superscript IV with ezDNase (Invitrogen). The *Ptch1*, *Gli1* and *Gapdh* primers were used previously (Li et al. 2014) (n=2-4 biological replicate per stage; error bars indicate SEM).

- The WB in Fig. 1C and Supp dataset 3 needs quantification. It seems that the level of both Gli3FL and Gli3R forms is higher in pre-HH stages.

The WB in Fig.1C is quantified and is in Supplementary Fig. 1d. Although trending, the higher levels of GLI3 in pre-HH limb buds are not significant. See new panel below.

D.

Supplementary Fig. 1D. Quantification of GLI3-FL and GLI3-R from western blot in Fig.1b (n=3; error bars indicate SEM).

How does the Gli3 expression level in the RNA-seq datasets compare between pre and post-hh stages? It would be much informative that the analysis at E10.5 is performed separately in the anterior and posterior bud, for a better comparison, as most Gli3R is anterior.

Gli3 expression from RNA-seq E9.25 and E10.5 anterior datasets is now depicted in Fig.S1B, which shows reduced *Gli3* expression in E9.25 compared to E10.5 limbs. Unfortunately, we did not generate RNA-seq from posterior limb buds. However, we agree that most *Gli3* is anterior, as *Gli3* mRNA is downregulated in the E10.5 posterior limb bud as a consequence of *Shh* signaling (Galli et al, 2010 PMID: 20386744).

- In the heatmap in Fig. 1G, the region marked as GBRs not bound pre-HH should correspond to 30%, however it seems smaller.

The brackets depicting the unbound 30% were incorrectly smaller than they should have been and have been corrected. We thank the reviewer for catching this error.

- It would be interesting to know the overlap between the GBRs not bound pre-HH (206/309) and those with called H3K27ac peaks (93/309).

Prior to HH signaling, only 39% (121/309) of HH-responsive GBRs have called H3K27ac peaks. Of those, 88%(107/121) are bound by GLI3 at that time. Therefore, most HH-responsive GBRs that have H3K27ac called peaks are bound by GLI3 in the pre-HH limb. We have added this to the results (page 6-7, line 125-133):

"In addition, only 39% (121/309) of HH-responsive GBRs have called H3K27ac peaks in the pre-HH limb, of which, 88% (107/121) are bound by GLI3 in the pre-HH limb, with only 14 GBRs not bound by GLI3 being enriched for H3K27ac at this time."

- My understanding is that most of the post-HH Gli repression refers to the anterior limb, as schematized in the last figure. I think that the spatial issue (anterior vs posterior) gets hidden along the paper and would suggest the authors to clarify this point. For example, does Fig. 5B refer to the whole limb bud?

Fig. 5B examines the posterior halves of limbs buds in order to compare posterior E10.5 WT limbs that have active HH signaling and low GLI repression, to posterior E10.5 *Shh*^{-/-} limbs that have no HH signaling and constitutive GLI repression. This allows us to then ask if GLI repression is established by E10, an intermediate timepoint, where we find it has not been fully established. We agree this did not come across well in the paper and have modified the manuscript, figure and figure legend to clarify this specific point. We have also worked to clarify whether anterior/posterior tissues were used for each experiment in figures, figure legends and the manuscript where appropriate.

- line 38: better Gli3R?

Line 38 and 42- "GLI-R" has been changed to "GLI3-R".

Minor points:

- Line 121: it is not clear to what these figures refer to (86% (6385/7382)). This has been clarified

"In contrast to HH-responsive GBRs, we previously found most GBRs remain stably acetylated in both the presence and absence of HH signaling at E10.5 (Lex et al. 2021). Consistent with this, 86% (6385/7382; Fig.S1e) of all E10.5 GBRs that are acetylated in E10.5 limb buds, have called H3K27ac peaks prior to HH induction."

- Personally, I would like to see the E10.5 acetylation heatmap of Fig. 2C also included in Fig. 3B, even is repeated, for a better comparison.

It would provide a useful comparison. However, we unfortunately used different ChIP protocols to generate these two datasets. The data in Fig 2 was generated using a conventional ChIP-seq protocol with 600k cells while we used a 'microChIP' protocol with 100k cells to generate the data in Fig. 3B. Because the data was not generated with the same approach, it isn't directly comparable. In addressing this comment, we realized that we did not indicate which datasets were derived from conventional or microChIP and have indicated when microChIP was used on figure legends and added details to the methods section.

- The *Hoxd12* and *Cdk6* GBS in Fig. 3e are not referred in the text.

We have added this information (page 8, lines 160-164):

"Additional GBRs around established *GLI3* target genes also did not show increased H3K27ac enrichment with loss of *Gli3*. This was evident at genes such as *Hoxd12* which lack WT E9.25 H3K27ac at GLI-bound regions and remain unchanged in *Gli3*^{-/-} limbs, or at genes like *Cdk6*, which has E9.25 acetylation, but does not gain further enrichment with loss of *Gli3* (Fig.3e)."

- CUT&Tag for K27me3 has been performed separately in the anterior and posterior E10.5 bud. If this is the case, the specific datasets used in the figures (Fig. 4G-H) should be indicated.

The dataset (Supplementary Dataset 1) for anterior and posterior H3K27me3 CUT&Tag is now listed in the manuscript when referencing Fig.4G-H. We have also added additional details in the figure legend.

- Because there is not an antibody that exclusively detects the Gli3R processed form, I would not make a strong point of the ciliary observations. Is there any concomitant change in the amount of Gli3 signal in the nucleus? Also, could the spike in the rate of ciliary Gli3 trafficking at E9.75, be detected in WBs?

As it is difficult to detect whether the spike in GLI ciliary trafficking results in an increase of GLI-R processing, we have removed the statements suggesting this possibility from the results section and only mention this during the Discussion. We haven't used WBs primarily because we suspect that the contribution from the ciliary pool would be drowned out by additional cytoplasmic GLI3.

- Supplementary Fig. 3D is not mentioned. The caption refers to Supplementary Fig. 3E.

We thank the reviewer for catching this. The figure legend has been corrected to include the description for Supplementary Fig. 3D and correctly label the Supplementary Fig. 3E description.

Reviewer #2 (Remarks to the Author):

In this manuscript Lex and collaborators study the function of Gli3 during limb bud formation in mice. Using different genomic techniques and different mutant mouse models, the authors clearly establish that the Gli protein does not act as a repressor of its target genes before the HH signal starts to function during limb development. Thus, the authors demonstrate an "inert" role of Gli in the early stages of limb development. This inert role is given because Gli is present but its loss does not increase the expression of the target genes nor the acetylation or accessibility of their enhancers. On the other hand, repression of target gene expression by Gli is established independently of HH signaling but only once HH signaling has begun. These results are of interest both for the knowledge of mouse limb development and patterning, and for the understanding of the function of the HH-Gli signaling pathway. For these reasons I think that this study can be accepted for publication in Nat Comm.

We thank the reviewer for their supportive comments.

I have a few minor comments that could improve or clarify some aspects of the manuscript.

- For readers who are not experts in the mouse model, the use of different terms to refer to developmental stages is quite confusing. Sometimes authors refer to them as E10, E10.5 etc and sometimes as a function of the number of somites. A homogenization of these terms in the manuscript could facilitate understanding. Personally, I prefer to use the number of somites as a reference, since the same stage (E10) may contain embryos with a greater or lesser number of somites.

We have worked to clarify somite stages when we reference an embryonic stage in the text and in figure legends, particularly in the sections describing the ATAC and ciliary experiments where we use E10 (28-30S and E9.75 (26-28S) limb buds.

- HH signalling activation is established by looking at Gli1 expression by in situ hybridization. This technique is not highly accurate and maybe RTqPCR could give a more precise moment for the HH activation.

We have taken this suggestion and performed qPCR on single forelimb bud pairs at somite stages spanning this developmental time-window, from 21-28S in addition to E10.5 (32-35S) WT and *Shh*^{-/-} controls. We have added this data as a new panel to Supplementary Fig.1c and refer to it on page 4 line 71. We also included primers for *Shh* but the conditions were not sensitive enough to detect it reliably at E9.5 levels on single forelimb pairs. See new figure below.

Supplementary Fig. 1C. Quantitative PCR showing expression of *Ptch1* and *Gli1* in forelimb pairs from individual embryos at specified somite stage. *Ptch1* and *Gli1* expression is normalized to *Gapdh*. RNA was isolated from individual forelimbs using Trizol and cDNA was generated using Superscript IV with ezDNase (Invitrogen). The *Ptch1*, *Gli1* and *Gapdh* primers were used previously (Li et al. 2014) (n=2-4 biological replicate per stage; error bars indicate SEM).

- In line 75, the authors say that Gli3-R is expressed at comparable levels at pre- and post-HH stages. From the Western blot presented it seems that postHH expression is lower than pre-HH expression. Would it be possible to shown a quantification of the Western blot?

The WB in Fig.1C is quantified and has been added to Supplementary Fig. 1d. Although trending, the higher levels of GLI3 in pre-HH limb buds are not significant.

- Upregulation of *Foxf1* and *Osr1* is explained as residual mRNAs from the lateral plate mesoderm. While this explanation seems convincing to me, would it be possible to check on scRNAseq data present in different databases to show that this genes expression is specific to the lateral plate mesoderm?

Foxf1 and *Osr1* are both considered signature genes of the lateral plate mesoderm (LPM) (Prummel et al, 2019 PMID# 31451684). *Foxf1* expression in the LPM is HH-dependent (Tsiairis and McMahon, 2009 PMID# 19879143). It is unknown if HH signaling regulates *Osr1* in the LPM, although *Osr1* is HH-dependent in anterior mesodermal tissues such as the foregut mesenchyme (Han et al, 2017 PMID# 28501478; Guzzetta et al, 2020 PMID# 32561646). Based on this, it seems reasonable that these genes are HH-dependent in the LPM. However, as exemplified by the latter example, we wish to emphasize that we do not think that these genes are not exclusively expressed in the lateral plate mesoderm and are expressed in other tissues throughout development. For example, *FoxF1* is expressed in multiple other tissue types, several of which have been also shown to be HH-dependent, including in the second heart field (Hoffmann et al, 2014 PMID# 25356765).

We examined the expression of both of these genes in scRNA-seq datasets (Cao et al. 2019) and while both of these genes are highly expressed in 'intermediate mesoderm', which likely encompasses lateral plate mesoderm, they are

also expressed in many other tissues (see Reviewer Figure 1 below). Although this data is consistent with our model, it doesn't shed additional information and so we would prefer to not include it in the actual manuscript.

Reviewer Figure 1. Expression of *Foxf1* and *Osr1* across cell types in mouse embryos. *Osr1* and *Foxf1* expression across cell types from scRNA-seq data sets from whole E9.5-E13.5 mouse embryos. Cell types depicted are top ~50% of cell types with detectable *Osr1* or *Foxf1* expression, across pooled data across all stages, E9.5-E13.5.

Cao J, Spielmann M, Qiu X, et al. The single-cell transcriptional landscape of mammalian organogenesis. *Nature*. 2019;566(7745):496-502. doi:10.1038/s41586-019-0969-x

- The authors use quantitative terms when talking about in situ hybridization experiments (ex. Lines 184-186). However, ISH is not quantitative. Would it be possible to quantify expression in such cases?

The majority of the data was acquired with conventional in situ hybridization (Fig. 4E), which we agree is not quantitative. Even the additional data we collected with HCR fluorescent in situ hybridization (Fig. 4D) is challenging to quantify in a wholemount embryo. One possible way to quantify this experimentally would be with quantitative RT-PCR for *Hand2* across anterior and posterior halves at the various stages and genotypes. However, the small size of the limb buds and the dynamic nature of the expression pattern would make this technically challenging. Since we cannot easily quantify gene expression, we have altered the language to emphasize that we are making relative comparisons within limb buds and that these are not quantitative.

“At early post-HH stages (24-26S), *Hand2* expression is variable in WT and *Gli3*^{-/-} limb buds, with most WT limb buds expressing anterior *Hand2* at levels comparable to those seen posteriorly. However, comparably lower levels of anterior *Hand2* expression were present even in some *Gli3*^{-/-} limb buds (Fig.4e, white arrow). Posterior restriction of *Hand2* is not evident until 26S and is still variable at that time (Fig.4e, Supplementary Fig.3e). Although not quantitative, these findings are consistent with previous reports that noted early co-expression of *Gli3* and *Hand2* at these stages¹⁴.”

- While there is no difference in Hand2 motif enrichment between pre-HH and post-HH when the authors studied the 310 Hand2 binding sites bound by Gli3 all together, would it be possible that specific sites in specific enhancers may be co-regulated?

This could be a possibility, most notably around genes that are bound by GLI3 at E9.25 but not E10.5. To address this, we have edited the following section on page 15, lines 374-380 to read:

"It is possible that other factors could be present at some ATAC-seq footprints in *Shh*^{-/-} limbs overlapping HAND2 motif regions, thus masking our ability to detect a loss of a HAND2 footprints. We also note that *Tbx2* and *Tbx3* are not reduced in pre-HH *Gli3*^{-/-} limb buds as would be expected if GLI3 had a significant role in co-activating these HAND2 target genes (Supplementary Dataset 4). While we cannot rule out the possibility of GLI3-HAND2 co-regulation at specific enhancers, we do not believe this to be a broad mechanism for initial regulation of GLI target genes."

- If Gli binding to GBRs in Pre-HH limb buds is inert, but these bound regions are enriched for poised enhancer modifications? Could we consider Gli as a pioneer factor?

This is an interesting point. We think that GLI3 binds more preferentially to poised regions rather than being responsible for helping poise them. We now discuss our thoughts on the role of GLI3 as a pioneer factor in the Discussion (pages 16-17, lines 399-406):

"Although GLI3 itself could act as a pioneer factor, we do not favor this possibility. First, while many regions that lack GLI3 binding also lack poised enhancer modifications, some regions remain accessible and poised, but are not yet bound by GLI3 at E9.25 (Fig.1f-i). This suggests that GLI3 binds preferentially to enhancers that are already poised, rather than playing a role in promoting deposition of these poised modifications. Additionally, while chromatin compaction of GBRs is GLI3-dependent, initially gaining accessibility is not, as we find that in E10.5 *Shh*^{-/-}/*Gli3*^{-/-} limbs, the chromatin is accessible, similar to E10.5 posterior WT controls (Fig.5)."

- Figure 6d the legend should say "Left, middle and right" instead of top middle and bottom.

The Fig.6d legend has been corrected to "left, middle, right".

Reviewer #3 (Remarks to the Author):

Shh regulates limb patterning largely by preventing formation of Gli3 repressor (Gli3R) and enabling de-repression of target genes, and cross-repression between high anterior Gli3R and posterior Hand2 is thought to play a critical role in "pre-patterning" limb A-P polarity prior to Shh activation. In this manuscript, Lex et al. use genome-wide approaches to systematically examine the role of Gli3R in the limb prior to initiation of Shh signaling (pre-HH), identifying Gli3 binding, chromatin status and expression of Shh limb target genes in WT, Gli3 KO, and Shh KO embryos at pre-Hh stage, compared with WT, Shh KO, or Shh/Gli3 KO (no activator, no repressor; similar to pre-Hh Gli3 KO condition) at post-Hh stage (after Shh activation). They present convincing evidence that even though a large proportion of targets are 'poised' for expression (accessible) and are Gli3-bound at pre-Hh stage, targets are surprisingly not repressed by Gli3 pre-Hh. Even Shh KO limb buds (with high Gli3R) appear not to have target repressive chromatin changes at a pre-Hh stage. In agreement, targets are not precociously de-repressed in the Gli3 KO; ectopic anterior activation in the Gli3 mutant commences post-Hh. On the other hand, a number of target genes do have PRC2 marks (H3K27me3), and/or "closed" chromatin, suggesting that other factors mediate pre-Hh repression. Focusing on pre-Hh Hand2 regulation, as a signature Gli3R target that is expressed pre-Hh as well as post-Hh, they show that Gli3 and Hand2 are extensively co-expressed pre-Hh and neither is strictly localized as proposed by the pre-patterning model. Although Gli3 is bound to Hand2 pre-Hh, it is functionally inert until the transition to post-Hh, which nevertheless occurs independent of Shh signaling (present in later stage Shh KO).

These results make a convincing case that Gli3 repression only becomes active following Shh activation and challenge the long-standing prepattern model for initiation of limb A-P polarity. Like other paradigm-shifting discoveries, this paper raises more questions than it answers: what regulates Hh-targets prior to pathway activation, and how is the change in Gli3R function triggered during this transition. The results likewise raise the possibility that similar regulation (lack of pre-Hh "competency" for Gli3R) applies in other contexts where the Shh pathway plays a major patterning role.

This work has potentially broad ramifications and will appeal to anyone interested in Hedgehog signaling. I have some suggestions below that the authors may wish to consider for improving the clarity and readability/accessibility of the paper.

We thank the reviewer for their positive evaluation and constructive suggestions.

-There is a great deal of (well-executed) data presented and summarized; it would be very helpful in digesting the data to know how particular 'signature' targets actually behave (which can't be inferred from Venn diagrams and heatmaps). The tabular summary of H3K27 methylation and acetylation profiles for the list of 74 high confidence targets (data set 5) is very useful in this regard; it would be even more helpful to add columns also summarizing the Gli3 binding status and chromatin accessibility status (open or closed) in the pre-Hh limb bud.

We agree and have added this information to Supplemental Dataset 5. It now includes columns for E9.25 GLI3 binding, H3K4me1, H3K4me2 and ATAC called peaks. Please see the first 15 rows of the excel file below, depicting these changes.

Gene	E9.25 GLI3	E9.25 H3K27me3	E9.25 H3K27ac	E9.25 H3K4me1	E9.25 H3K4me2	E9.25 ATAC
Hoxd12	YES	YES	NO	NO	NO	YES
Hoxd13	YES	YES	NO	YES	YES	YES
Ptch1	YES	YES	YES	YES	YES	YES
Hoxd11	NO	YES	NO	YES	NO	YES
Hoxd10	YES	YES	YES	YES	YES	YES
Sall3	YES	YES	YES	YES	YES	YES
Hsd11b2	NO	YES	YES	YES	YES	YES
Socs2	YES	YES	YES	YES	YES	YES
Aprt	YES	NO	YES	YES	YES	YES
Mfsd2a	YES	YES	NO	YES	YES	YES
Foxd2	YES	YES	NO	YES	YES	YES
Tagln2	YES	YES	NO	YES	YES	YES
Ccnd1	YES	YES	YES	YES	YES	YES
Cntfr	NO	YES	YES	YES	YES	YES

-Related to the above point, for different types of analyses, peak profiles for different gene target examples are often chosen. Much of the detail understandably focuses on Hand2, which is distinctive in already being highly expressed pre-Hh. Likewise, Ptch1 and Gli1 are a special subclass that is uniquely GliA dependent. It would be nice to also show the complete set of peak profiles (Gli3, ATAC, H3K27me3, H3K27Ac, in WT and in mutants) for a couple of selected high-confidence targets that are not highly expressed pre-Hh but are Gli3R-regulated after Shh activation, and include the major different sub-classes in data set 5 (for eg. Grem1, Hoxd13, Hhip, Ets2). This would highlight the differences among subclasses more clearly. For example, Grem1 has closed chromatin early, yet surprisingly also has appreciable H3K27Ac; presumably the reverse is the case for some other targets such as Hoxd13.

We agree with this suggestion. As this is a lot of data to show in one figure, we have added this as two additional supplementary figures. These figures show genes that are likely incompetent to GLI repression (either not bound by GLI3 at E9.25 or have high H3K27me3; Supplementary Fig.7 – see below) and genes that are likely competent to be repressed by GLI3 at E9.25 (bound, poised and lack H3K27me3; Supplementary Fig.8 – see below).

Supplementary Fig.7. GLI3 target genes not competent for GLI repression. A. Enhancer profiles of the HH target gene *Gremlin* and *GRE1*, the GBR regulating *Gremlin*. *GRE1* is not bound by *GLI3* prior to HH signaling and is not poised or accessible, while the promoter of *Gremlin* has high *H3K27me3* enrichment, and thus is not competent for *GLI* repression. B. Example of GBRs that are not bound but are poised, near HH target *Smoc1*. bound that are not bound by *GLI3* prior to HH signaling at E9.25 and are not competent for *GLI3* repression. C. Examples of GBRs at genes that are bound by *GLI3* and poised, but have high *H3K27me3* and are likely not competent for *GLI3* repression. Differences in *H3K27ac* at gene promoters may result in slow (left, middle) or fast (right) gene activation. Scale bars = 1kb.

Supplementary Fig. 8. GLI3 targets competent for GLI repression. A. Examples of GBRs bound by GLI3 at E9.25, that are poised and lack H3K27me3 and should be competent for GLI3 repression prior to HH. B. Example of a GBR that is poised and bound by GLI3 specifically at E9.25, representing a potential stage-specific GLI regulated target. Scale bars =1kb.

The authors argue that Gli2, which can have a weak repressor activity in certain contexts, does not likely compensate for the apparent lack of Gli3R function at pre-Hh stage. Some of these arguments are more convincing than others. --Whether the Gli2/3 double KO has no further digit patterning phenotype is debatable; at early-stage removal, digit number is certainly increased beyond Gli3 KO alone and "patterning"/morphology changes per se are more difficult to assess, considering other late (Ihh) effects.

The argument that Gli2R function isn't evident in *Shh*/Gli3KO limbs (which are similar to the Gli3 KO) assumes that Gli2R function is also not evident in the Gli3 KO, but this is not really known (the Gli2/3 KO has not been analyzed in any detail in terms of early target gene expression compared to Gli3 KO).

Finally, the authors do note that even in Gli3R-regulated targets post-Hh, compaction due to Gli3R seems to occur later, after the onset of actual repression by Gli3R. Together with the other evidence in this paper that repression and compaction are not always synonymous (PRC2+ and open chromatin on some targets pre-Hh), this raises a caveat for concluding absence of both Gli2R and Gli3R function.

We acknowledge the reviewers point here and, upon reflection, agree that we didn't fully consider possible roles for Gli2R. We have revised the relevant discussion to discuss this caveat accordingly (edits are underlined):

"Possibility of GLI2 serving as a functionally redundant repressor in the absence of GLI3"

One possible explanation for the lack of GLI3 repressor activity is functional redundancy with other GLI proteins. While *Gli1* is not expressed in pre-HH limb buds (Fig.1b, Supplementary Fig.1a,b), GLI2 is present (Supplementary Fig.5c,e). GLI2 acts as a transcriptional activator in most embryonic tissues⁵⁸⁻⁶¹ but in certain contexts has weak repressor properties^{4,58,62-64}. Thus GLI2-R could potentially compensate for a loss of GLI3-R in a fashion that is specific to pre-HH limb buds that might subsequently become dependent on GLI3-R alone. Although *Gli2^{-/-}*; *Gli3^{c/c}* hindlimb skeletal phenotypes have been reported⁶⁵, they have not been examined at pre-patterning stages. Similarly, although the possibility that loss of GLI2-R might further enhance phenotypes in *Shh*; *Gli3* double mutants is unknown. On the other hand, in pre-HH WT limb buds, the GLI3-R target *Hand2* is initially co-expressed with *Gli3* throughout the entire limb bud before becoming posteriorly restricted by GLI3 in later limb development¹⁴ (Fig.4e; Supplementary Fig.3e). At least in these WT pre-HH limbs, GLI2-R does not function to prematurely repress *Hand2*, one of the signature repression targets for GLI3."

-In the discussion, the authors speculate on how targets may be repressed pre-Hh, which could be mediated by

PRC2 for those with H3K27 methylation or bivalent marks, or may occur because pioneer factors are required to activate those that have inaccessible, closed chromatin. But a third class remains, as the authors note, that has open chromatin and is not PRC2-regulated, and yet is not highly expressed pre-Hh. How do they envision that this class is regulated?

We have added several sentences describing how we think this subgroup is regulated in the Discussion:

“As this group of genes is not highly expressed in the early limb, and is not repressed by GLI3, we predict that HH-independent mechanisms may be required to activate them prior to the establishment of GLI3 repression. Most genes that are regulated by GLI3 in the post-HH limb are already expressed in E9.25 limb buds (Supplementary Fig.3d), suggesting that genes might have to become activated prior to GLI3 repressing them.

minor comments:

Suppl Figure 3 -- the legend is missing a description for panel D, and the panel E image is described as D in the legend.

The figure legend has been corrected to include the description for Supplementary Fig. 3D and correctly label the Supplementary Fig. 3E description.

line 378-9: sentence is poorly phrased (unclear pronoun reference).

This sentence has been revised:

“As HDACs require co-repressor complexes to guide them to their substrates⁵⁶, the absence of a functional GLI3 co-repressor might prevent HDACs from properly regulating GLI3 enhancers in the early limb.”

Reviewers' Comments:

Reviewer #1:

Remarks to the Author:

The authors have appropriately addressed my previous comments, I have no further concern, just very minor comments (see below).

The revision was a little confusing that the pages/lines indicated for the changes did not correspond to the final manuscript (although they also provided a version with tracked changes that did not include the figures). I would ask the authors to carefully go through their final version of the manuscript to fix some mistakes.

very minor comments:

- The caption for Supplementary Fig. 1C does not include the method as in the rebuttal. I think the RT-qPCR method is missing in the manuscript. Also, the reference to former Supplementary Fig. 1C in the caption of Fig. 4 should be changed to Supplementary Fig. 1E.
- The authors indicate that the quantification in Supplementary Fig. 1D corresponds to the WB in Fig. 1B (that in the new version is 1C). However, shouldn't it correspond to the quantification of the three WB shown in Supplementary dataset 3?
- I agree with the authors' explanation for not including the E10.5 acetylation map in Fig. 3b. Just out of curiosity, I understand that the pre-HH acetylation maps in Fig. 2C and in Fig. 3B, which look identical to me, were obtained with different ChIP protocols (CUT&RUN and CUT&tag respectively).

Reviewer #2:

Remarks to the Author:

The authors have convincingly answered all the questions and criticisms I made in my review. They have also added some experiments that clarify some of the points I discussed. I believe that the addition of these experiments and the comments and modifications to the text improve the initial manuscript, for which I already said I considered it suitable for publication in Nat Comm. I therefore accept the paper for publication in its present format.

Reviewer #3:

Remarks to the Author:

I think that the authors have addressed all of the reviewers' comments and improved on what was already an excellent paper. I believe this work will be of great interest and should be published.

Two of the reviewers had no further comments. The third reviewer identified missing or incorrect information listed in the manuscript or figure legends. Please see below for a point by point response of the reviewer's comments, explaining the corresponding corrections made in the manuscript or figure legends.

-The caption for Supplementary Fig. 1C does not include the method as in the rebuttal. I think the RT-qPCR method is missing in the manuscript. Also, the reference to former Supplementary Fig. 1C in the caption of Fig. 4 should be changed to Supplementary Fig. 1E.

The method, as stated in the rebuttal, is now correctly included in the Supplementary Figure Legend and in the manuscript Methods. Reference to Supplementary Fig. 1e (formerly 1c), has been correctly updated in the figure legends.

-The authors indicate that the quantification in Supplementary Fig. 1D corresponds to the WB in Fig. 1B (that in the new version is 1C). However, shouldn't correspond to the quantification of the three WB shown in Supplementary dataset 3?

We agree the wording in the legend did not accurately depict the actual quantification. The figure legends has been updated to state:

"D. Quantification of GLI3FL and GLI3-R from representative western blot in Fig.1c and additional western blot replicates in Source Data"